



# Comparison of dust optical depth from multi-sensor products and the MONARCH dust reanalysis over Northern Africa, the Middle East and Europe

Michail Mytilinaios[1], Sara Basart[2], Sergio Ciamprone[1], Juan Cuesta[3], Claudio Dema[1], Enza Di Tomaso[2], Paola Formenti[4], Antonis Gkikas[5], Oriol Jorba[2], Ralph Kahn[6], Carlos Pérez García-Pando[2,7], Serena Trippetta[1], and Lucia Mona[1]

[1]Consiglio Nazionale delle Ricerche-Istituto di Metodologie per l'Analisi Ambientale (CNR-IMAA), Tito Scalo, Italy
[2]Barcelona Supercomputing Center (BSC), Barcelona, Spain
[3]Univ Paris Est Creteil and Université Paris Cité, CNRS, LISA, F-94010 Créteil, France
[4]Université Paris Cité and Univ Paris Est Creteil, CNRS, LISA, F-75013 Paris, France
[5]National Observatory of Athens-Institute for Astronomy, Astrophysics, Space Applications and Remote Sensing (NOA-IAASARS), Penteli, Greece
[6]Earth Sciences Division, NASA Goddard Space Flight Center, Greenbelt, Maryland, USA
[7]Catalan Institution for Research and Advanced Studies (ICREA), Barcelona, Spain

**Correspondence:** Michail Mytilinaios (michalis.mytilinaios@imaa.cnr.it)

**Abstract.** Aerosol reanalysis datasets are model-based observationally constrained continuous 3D aerosol fields with relatively high temporal frequency that can be used to assess aerosol variations and trends, climate effects and impacts upon socio–economic sectors, such as health. Here we compare and assess the recently published MONARCH high resolution regional desert dust reanalysis over Northern Africa, the Middle East and Europe (NAMEE) with a combination of ground-based observations and space-based dust retrievals and products. In particular, we compare the total and coarse dust optical depth (DOD) from the new reanalysis with DOD products derived from MODIS, MISR and IASI space-borne instruments. Despite the larger uncertainties, satellite-based datasets provide a better geographical coverage than ground-based observations, and the use of different retrievals and products allows for at least partially overcoming some single-product weaknesses in the comparison. Nevertheless, limitations and uncertainties due to the type of sensor, its operating principle, its sensitivity, its temporal and spatial resolution, and the methodology for retrieving or further deriving dust products, are factors that bias the reanalysis assessment. We, therefore, also used ground-based DOD observations provided by 238 stations of the AERONET network located within the NAMEE region as a reference evaluation dataset. In particular, prior to the reanalysis assessment, the satellite datasets were evaluated against AERONET, showing moderate underestimations in the vicinities of dust sources and downwind regions, whereas small or significant overestimations, depending on the dataset, can be found in the remote regions. Taking into consideration these results, the MONARCH reanalysis assessment showed that total and coarse DOD simulations are consistent with satellite and ground-based data, capturing qualitatively the major dust sources in the area as well as the dust transport patterns. Moreover, the reanalysis reproduces the seasonal dust cycle, identifying the increased dust activity occurred in the NAMEE region during spring and summer. The quantitative comparison between the MONARCH reanalysis DOD and satellite multi-sensor products shows that the reanalysis tends to slightly overestimate the desert dust that





is emitted from the source regions and underestimate the transported dust over the outflow regions, implying that the model removal of dust particles from the atmosphere, through deposition processes, is too effective. More specifically, small positive biases were found over the Sahara Desert (0.04) and negative biases over the Atlantic Ocean and the Arabian Sea (−0.04), which constitute the main pathways of the long-range dust transport. Considering the DOD values recorded on average there, such discrepancies can be considered low as the low relative bias in the Sahara Desert (< 0.5) and over the adjacent maritime

regions (< 1), certifies. Similarly, over areas with intense dust activity the linear correlation coefficient between the reanalysis simulations and the ensemble of the satellite products is significantly high for both total and coarse DOD, reaching 0.8 over the Middle East, the Atlantic Ocean and the Arabian Sea, and exceeding it over the African continent. Moreover, the low relative biases and high correlations are associated with regions where large amounts of observations are available, allowing for robust model assessment.

## 1  Introduction

Atmospheric desert dust is one of the major contributors to global aerosol loading and is the dominant component of atmospheric aerosols over large areas of Earth (Zender et al., 2004; Goudie and Middleton, 2006) with the Sahara Desert as the main contributor to the aerosol budget at global scale (Middleton and Goudie, 2001; Prospero et al., 2002; Ginoux et al., 2012a). Mineral dust particles, suspended in the atmosphere from arid and semi-arid regions, can remain aloft for periods ranging from

several days to about a week, depending on their size (Prospero, 1999). Huge amounts of dust can be transported over great distances under favorable meteorological conditions, affecting regions hundreds to thousands of kilometers away (Mona et al., 2006; Papayannis et al., 2008, 2014; Flaounas et al., 2015; Gkikas et al., 2015; Ramaswamy et al., 2017; Yu et al., 2021).

The impact of atmospheric dust on the environment, health, and economies represents a major scientific and societal issue (UNCCD, 2022). Dust aerosols can interact with solar and thermal radiation and with clouds, affecting radiative forcing and

precipitation formation and thus influencing Earth's weather and climate (Levin et al., 1996; Tegen et al., 1996; Myhre and Stordal, 2001; Slingo et al., 2006; Lambert et al., 2013; Myhre et al., 2013; Nabat et al., 2015; Karydis et al., 2017; Gkikas et al., 2018, 2019). Once the dust is deposited, by wet or dry deposition, it impacts both aquatic and terrestrial ecological systems through their biogeochemistry (Okin et al., 2004; Jickells et al., 2005; Painter et al., 2007; Lekunberri et al., 2010; Yu et al., 2015). For countries in and downwind of arid regions, airborne sand and dust pose a significant threat to human

and animal health (Gyan et al., 2005; Griffin, 2007; Kanatani et al., 2010; Mallone et al., 2011; Cadelis et al., 2014; Pérez García-Pando et al., 2014; Querol et al., 2019; WHO, 2021) and to various socio–economic sectors, such as aviation, ground transportation, agriculture, infrastructure, solar energy and other industries (Goossens and Van Kerschaever, 1999; Sivakumar, 2005; Stefanski and Sivakumar, 2009; Mani and Pillai, 2010; Jiang et al., 2011; Weinzierl et al., 2012; Lekas et al., 2014; Costa et al., 2016; Al-Hemoud et al., 2017; Middleton, 2017; Kosmopoulos et al., 2018; Miri and Middleton, 2022; Monteiro et al.,

2022). It is therefore of great societal and scientific interest to better understand atmospheric dust processes, predict dust events and prevent or mitigate their unwanted impacts where possible.





A key parameter for tracking the airborne aerosols (including mineral dust) from satellite platforms and ground-based remote-sensing networks is aerosol optical depth (AOD). AOD is a quantitative measure of the attenuation of light as it is transmitted through the atmosphere, due to scattering and absorption by aerosols. As a result, AOD is proportional to the to-

tal amount of aerosol particles suspended in the atmosphere, providing important information about their concentration and variability; AOD spectral dependence is related to column-effective size distribution. Accordingly, coarse AOD is the fraction of the total AOD associated with coarse aerosol particles (i.e., larger than 0.5 μm in radius) in the atmosphere, and it is dominated by natural aerosols (i.e., sea salt and mineral dust; Carslaw et al., 2010). AOD wavelength-dependence is related to particle size, which has implications for climate, as direct radiative forcing induced by atmospheric aerosols depends strongly

on particle size. Accordingly, studies suggest that fine dust generally produces cooling whereas coarse dust tends to produce warming (Tegen and Lacis, 1996; Miller et al., 2006; Mahowald et al., 2014; Kok et al., 2017), although there remains significant uncertainty in mineral dust properties and therefore their impact on climate projections (Myhre et al., 2013, see Fig. 8.17).

Over the last two decades, satellite and ground-based sensors have made systematic aerosol observations on a global scale,

facilitating the integrated study of atmospheric aerosols and combining various measurement techniques and data analysis methods. Moreover, technological advancements nowadays allow for more detailed aerosol characterization, such as the estimation of mineral dust particle contributions to measured optical properties, providing an improved depiction of the atmospheric dust distribution globally (Kaufman et al., 2005; Liu et al., 2008, 2018; Giles et al., 2012; Peyridieu et al., 2013; Kahn and Gaitley, 2015; Gkikas et al., 2013, 2016; Marinou et al., 2017; Proestakis et al., 2018). Nevertheless, there are limitations

regarding the spatiotemporal coverage of aerosol observations and aerosol typing. Ground-based measurements may provide high sampling frequency, e.g., one or more measurements per hour; however, they are limited to over-land surfaces and provide very limited spatial coverage. Further, the distribution of surface stations is not ideal in itself for studying the highly varying desert dust concentrations, and the regions most affected by sand and dust storms are generally not well supported by research infrastructures and networks (Benedetti et al., 2018). On the other hand, polar-orbiting satellite sampling capabilities above

both land and sea are also limited, due to lower temporal resolution, as they obtain global coverage at best every 1 to 2 days (e.g., MODIS). For both surface and space-based aerosol remote sensing, data availability is affected by weather conditions (e.g., clouds and snow), and instruments that observe reflected or transmitted solar radiation (e.g., MODIS, MISR, AERONET sun-photometers) cannot obtain measurements during nighttime. Additionally, there is no single "best" aerosol satellite product globally, e.g., some large differences are observed when comparing products from different sensors and algorithms (Sogacheva

et al., 2020).

To fill these gaps, and overcome sparse coverage, low temporal resolution, and partial information provided by measurements, model simulations can be combined with observations within a data assimilation framework to estimate optimally the initial conditions for forecast models (analyses) and for the production of reanalysis datasets, i.e., complete and consistent reconstructions of the atmosphere. Aerosol reanalysis datasets can accurately represent the spatial and temporal distribution of

airborne dust over an extended period of time (Inness et al., 2013, 2019; Cuevas et al., 2015; Lynch et al., 2016; Gelaro et al., 2017; Yumimoto et al., 2017), reducing the estimated errors in numerical model simulations due to imperfect model dynamics





as well as to uncertainties in the initial conditions and forcing fields, by means of assimilated observational constraint. A novel regional reanalysis of desert dust aerosol over the Northern Africa, the Middle East, and Europe (NAMEE) domain has been released recently by the Barcelona Supercomputing Center (Di Tomaso et al., 2021) for the period 2007–2016. The reanal-
ysis was obtained using the MONARCH aerosol–chemical weather system and by assimilating a satellite AOD dataset that specifically constrains the dust component. The MONARCH dust reanalysis aims to provide reliable dust information at high temporal and spatial resolution, both near the surface and at upper levels. The reanalysis dataset consists of three-dimensional (3D) and two-dimensional (2D) variables covering a wide range of dust-related atmospheric parameters, including optical and microphysical dust properties along with dust deposition and solar radiation variables. Di Tomaso et al. (2022) describes the
reanalysis set-up as well as data assimilation diagnostics and provides a first basic evaluation of the reanalysis.

Here, we present a comprehensive assessment of the MONARCH reanalysis total and coarse mode dust optical depth (i.e., DOD and coarse DOD, respectively) at 550 nm against satellite-based mineral dust products retrieved or derived from different sensors (i.e., MODIS, MISR and IASI), along with ground-based AERONET AOD measurements. DOD is the model diagnostic variable directly constrained by observations through data assimilation and, therefore, is the primary focus of our reanalysis
assessment. The validation of variables that are not directly constrained by observations such as the vertical extinction profile will be the subject of a companion study. Rather than use a single DOD reference dataset, we combine different DOD products that together provide better coverage of the model's spatiotemporal domain. An additional advantage of using different observational reference datasets is the ability to perform cross-validation of model performance, based on the results obtained from each dataset. The total and coarse DOD products of the reference datasets were obtained following different retrieval tech-
niques and assumptions; limitations of each dust characterization technique introduce uncertainties into the DOD retrievals. Nevertheless, by collating the comparison results obtained from different datasets, we can identify biases caused by retrieval uncertainties, and consider them in the final reanalysis assessment. To further investigate the reliability of the satellite-based DOD datasets, we also evaluated all products using an independent observational dataset (i.e., AERONET) as reference.

The following sections describe the assessment process and the results obtained. In Sect. 2 we present the main charac-
teristics of the datasets used for the assessment of the MONARCH DOD reanalysis along with a description of the applied methodology. In Sect. 3, validation of the satellite data using AERONET ground-based measurements is presented. Results from the MONARCH reanalysis assessment procedure are presented in Sect. 4, whereas in Sect. 5 the main findings and conclusions are summarized.

## 2 Datasets and methodology

The dust-related observational datasets selected for the MONARCH dust regional reanalysis assessment include remote-sensing products from ground-based networks (i.e., AERONET) and satellite sensors (i.e., MODIS, MIRS, and IASI). The selection of these remote-sensing-derived dust products considers the following requirements: i) the observational datasets should have sufficient temporal and geographical coverage over the reanalysis dataset (i.e., NAMEE region and the period 2007–2016); ii) datasets must be consolidated in order to assure good quality data and to assess the associated errors; iii) the





data should be harmonized in terms of procedures and quality control within the specific dataset; lastly, iv) dust speciation is essential for the dust reanalysis assessment. The latter means that the aerosol observational products should be related not to the total AOD, but specifically to its dust component, obtained through advanced products or through consolidated dust-filtering algorithms. Finally, in the assessment, it is important to consider that the observational and modeling datasets are usually available at different spatial and temporal resolutions, which implies that they must be collocated in terms of space and time before

their comparison. Details about the dust AOD characterization and the spatiotemporal collocation methodology followed for every dataset are given in the next subsections.

## 2.1    MONARCH dust regional reanalysis

The MONARCH dust regional reanalysis represents the state of the art desert dust information over a domain covering the most prominent dust source areas in Northern Africa and the Middle East. This dataset has recently been released by the Barcelona

Supercomputing Center (BSC) for a 10-year period, spanning from 2007 to 2016, over a spatial domain extending from $0°$ N to $70°$ N latitude and from $-30°$ E to $70°$ E longitude. An extensive description of the reanalysis set-up and dataset can be found in Di Tomaso et al. (2022). Here we summarize the main characteristics that are relevant for this study. The MONARCH reanalysis geographical domain includes some of the world's main dust sources like the Sahara in Northern Africa, the Arabian Desert in the Middle East and the arid regions of Western Asia (Fig. 1), with the former emitting 50 % of total dust burden in

the atmosphere (Ginoux et al., 2012b). It also includes maritime regions such as the Arabian Sea, the Mediterranean Sea and the northeastern Atlantic Ocean, over which long-range dust transport takes place frequently. A list of desert and arid regions, representing the major dust sources of the NAMEE region, is denoted by capital letters in Fig. 1. Figure 1 also shows the 10 sub-regions in which the MONARCH reanalysis domain is divided for evaluation purposes.

     MONARCH reanalysis novelty includes its unprecedented spatial and temporal resolution, as well as the assimilation of

an innovative DOD dataset covering all cloud-free and snow-free land surfaces, including areas particularly relevant for dust applications such as very bright reflective surfaces. Reanalysis fields are available at a 3-hourly time-step (starting every day at 03:00 UTC) and at a horizontal resolution of $0.1°$ latitude $\times$ $0.1°$ longitude in a rotated grid ($\sim 10\,\mathrm{km} \times 10\,\mathrm{km}$ at the Equator). The reanalysis has been obtained using the Multiscale Online Non-hydrostatic AtmospheRe CHemistry model (MONARCH; Pérez et al., 2011; Klose et al., 2021) and satellite coarse-mode DOD at 550 nm derived from the MODerate resolution Imaging

Spectroradiometer (MODIS) instrument, operating aboard NASA's Aqua satellite. More specifically the dataset assimilated in the MONARCH reanalysis consists of gridded coarse DOD retrievals over land surfaces, including desert areas, derived from the MODIS Deep Blue aerosol products (Collection 6, Level 2; Hsu et al., 2004) according to the retrieval procedure described in Ginoux et al. (2010, 2012a) and Pu and Ginoux (2016). Data assimilation was performed by means of a local ensemble transform Kalman filter data assimilation scheme with a four-dimensional extension (Hunt et al., 2007; Miyoshi and Yamane,

2007; Schutgens et al., 2010; Di Tomaso et al., 2017; Tsikerdekis et al., 2021; Escribano et al., 2022).

     The reanalysis dataset consists of upper-air variables such as dust mass concentration and extinction coefficient at 550 nm, surface fields such as accumulated dust dry and wet deposition and mass surface concentration, and total column fields like instantaneous total column dust load, DOD and coarse DOD at 550 nm. Calculation of basic ensemble statistics was performed





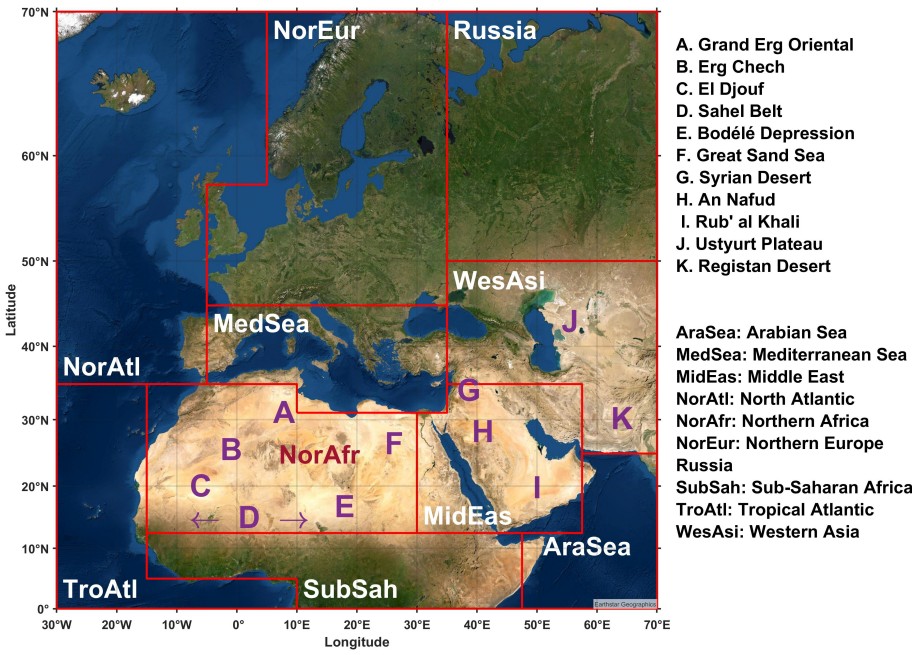

**Figure 1.** MONARCH reanalysis geographical domain (base map source: Esri, Earthstar Geographics, CNES/Airbus DS). The domain is divided into ten sub-regions; capital letters in purple mark the major deserts in Northern Africa, the Middle East and the Western Asia.

for each reanalysis variable, namely the ensemble arithmetic mean, standard deviation, median and maximum; however, in this paper, we assess exclusively the reanalysis mean, as it is a more representative value than the median for describing the ensemble, as it considers all the members of the ensemble without excluding the outliers.

MONARCH follows a sectional approach for atmospheric dust, i.e., the size distribution is decomposed into eight size bins, corresponding to different dust particle ranges with particle radius ranging from 0.1 µm (fine particles) to 10 µm (coarse particles). The MONARCH reanalysis DOD is produced considering all eight model size bins, whereas the coarse mode DOD includes the five coarser size bins from 0.6 to 10 µm in dust particle radius. For simplicity, hereafter, we refer to the MONARCH reanalysis also as MONARCH.

### 2.2 MODIS-based dust product: MIDAS

The MODIS total and coarse DOD used in this study is based on the recently developed ModIs Dust AeroSol (MIDAS) dataset (Gkikas et al., 2020, 2021). MIDAS combines quality filtered AOD from MODIS (on board NASA's Aqua satellite) at swath level (Collection 6.1, Level 2), along with DOD-to-AOD ratios provided by the Modern-Era Retrospective analysis for Research and Applications version 2 (MERRA-2) reanalysis (Gelaro et al., 2017) to calculate the contribution of mineral dust particles to the overall AOD on the MODIS native grid. MIDAS coarse mode DOD is also derived using the MERRA-2 DOD fraction, and considers only particles with radius larger than 0.5 µm. MIDAS provides columnar daily total and coarse





DOD (at 550 nm) over all cloud-free and snow-free land and ocean surfaces, at fine spatial resolution (0.1° × 0.1°), and over a
15-year period (2003–2017).

The uncertainty of the MIDAS DOD dataset was estimated by taking into account the uncertainties of the MODIS AOD and MERRA-2 DOD-to-AOD ratio (Gkikas et al., 2021), which in turn were calculated by using AERONET AOD (Giles et al., 2019) and LIVAS (Amiridis et al., 2015; Marinou et al., 2017) dust fraction, respectively, as a reference. According to the uncertainty analysis performed, MIDAS uncertainties scale with DOD value (Gkikas et al., 2021, see Fig. 8); however, in
terms of relative uncertainty the MIDAS DOD product is highly reliable over dust-rich regions and becomes more uncertain in areas where dust loading is infrequent. Although the MIDAS coarse DOD product is still under testing, it was used in this study after being evaluated against ground-based AERONET coarse DOD observations (Sect. 3).

Prior to the comparison, the MONARCH and MIDAS datasets were collocated in space and in time. First, MONARCH was re-gridded through bilinear interpolation, using the MIDAS grid as a reference. Regarding the temporal collocation, thanks
to the wide MODIS swath ($\sim 2330$ km), MIDAS provides near-global DOD retrievals every 1 to 2 days; consequently, MONARCH 3-hourly time-steps had to be averaged around Aqua's overpass time. Aqua follows a sun-synchronous, near-polar orbit, crossing the Equator once during daytime at $\sim 13$:30 local time (LT). As MONARCH outputs are given in UTC, it was necessary to convert 13:30 LT to UTC units, which depends on longitude. The MONARCH spatial domain contains eight time-zones (15 degrees of longitude constitute one time-zone) from $-2$ hours to $+5$ hours, implying that Aqua/MODIS takes
measurements over the MONARCH domain between 08:30 UTC (70° E) and 15:30 UTC ($-30°$ E). Hence, for a given longitude (assume 43° E) related to a certain time-zone (i.e., $+3$ hours), the MONARCH DOD was temporally averaged around MODIS acquisition time (i.e., 10:30 UTC) using the two nearest MONARCH timeslots (i.e., 09:00 and 12:00 UTC).

## 2.3 MISR dust product

Multi-angle Imaging SpectroRadiometer (MISR) is an imaging instrument, which provides aerosol observations on a global
scale since 2000 (Diner et al., 1998). The MISR instrument consists of nine cameras observing at nine different view angles (between $-70.5°$ and $70.5°$), and in four different wavelengths (446.4 nm, 557.5 nm, 671.7 nm and 866.4 nm). Apart from the AOD retrievals in the four spectral bands, the variations between the reflectance acquired from a very large range of scattering angles can provide information about aerosol microphysical properties such as particle size, shape, and single-scattering albedo by considering the appropriate particle optical models (Kahn et al., 1998, 2001; Kahn and Gaitley, 2015). In particular, MISR's
sensitivity to the characteristics of the aerosol scattering phase function enables it to distinguish between the non-spherical and spherical particles, making it possible to separate mineral dust aerosols from other aerosol components (Kahn et al., 1997). Thus, the AOD fraction of the non-spherical particles, consisting of randomly oriented non-spherical grains or ellipsoids, can be considered equivalent to the DOD with relative certainty, especially over dark-water surfaces (Kalashnikova and Kahn, 2006). Generally, like nearly all passive satellite aerosol remote-sensing, MISR AOD retrievals are less reliable over bright surfaces
(Kahn et al., 2010); therefore, for the MONARCH DOD comparison we used exclusively dark-water retrievals, which are exceedingly sensitive to aerosol non-sphericity (Guo et al., 2013; Kalashnikova et al., 2013). In particular, we used the daily





dark-water non-spherical AOD retrieval (at 557.5 nm) provided by the MISR Level 3 Component Global Aerosol Product (MIL3DAE, Version F15_0031) dataset, on a $0.5° \times 0.5°$ spatial grids during the period 2000–2016.

However, we should note here that the dark-water retrieval sensitivity to particle non-sphericity decreases when the total AOD is below about 0.1 and when the non-spherical component contributes less than 15–20 % to the total AOD (Kalashnikova and Kahn, 2006; Pierce et al., 2010; Kalashnikova et al., 2013). As a result, non-spherical particles are sometimes retrieved over remote oceans, even where they are unlikely to be present, overestimating non-spherical AOD fraction, probably due to the presence of unscreened cirrus or other naturally occurring non-spherical aerosols (Pierce et al., 2010; Kalashnikova et al., 2013; Kahn and Gaitley, 2015). On the other hand, the dark-water non-spherical AOD retrieval performs quite well in regions

of dust transport where the AOD values are significant and the non-spherical component is dominant. As previous studies have shown (Kalashnikova and Kahn, 2006, 2008), this is especially true over the Tropical Atlantic where desert dust is the dominant aerosol component, accounting for 40–70 % of the total AOD (Guo et al., 2013).

The spatial collocation between the two datasets was obtained by re-gridding MONARCH DOD, using the coarser MISR Level 3 product grid as a reference. For the temporal collocation we followed a similar methodology as in the case of MIDAS:

MISR on board of NASA's Terra satellite is crossing the equator on the descending node at about 10:30 LT. After having converted 10:30 LT to UTC time taking into account the related longitude, the MONARCH DOD was temporally averaged around MISR overpass time using the two nearest MONARCH timeslots. MISR has only 1/4 to 1/3 the spatial sampling of MODIS due to its relatively narrow swath width ($\sim$ 380 km), resulting in global coverage every 7–9 days at mid-to-low latitudes, compared with 1–2 days by MODIS. So, sampling must be taken into consideration when comparing datasets

averaged over longer timescales.

## 2.4 IASI dust product: AEROIASI

The Infrared Atmospheric Sounding Interferometer (IASI) instrument is in orbit onboard EUMETSAT's MetOp satellite, providing temperature and water vapor profiles of the troposphere and lower stratosphere at vertical and horizontal resolutions of 1 km and 12 km, respectively. IASI measurements in the infrared part of the electromagnetic spectrum enable observations

both in daytime and nighttime conditions. Thanks to its wide swath (2200 km), IASI provides global coverage twice a day, crossing the equator on the descending node at approximately 09:30 and 21:30 LT. Desert dust profiles can be derived from individual thermal infrared spectra measured by IASI for most cloud-free IASI pixels, both over land and ocean, following the method called AEROIASI, developed by Cuesta et al. (2015). Information on the vertical distribution of dust is provided mainly by their broadband radiative effect, which includes aerosol thermal emission depending at each altitude on the vertical

profile of temperature. AEROIASI products include twice-daily 3D distributions of dust extinction coefficient, although the present study only uses dust horizontal distributions derived in terms of DOD.

The AEROIASI algorithm firstly uses as input a priori dust microphysical properties (e.g., a dust number concentration profile as well as its size distribution and complex refractive index) and meteorological variables (temperature profiles, surface temperatures and $H_2O$ profiles) to simulate thermal infrared radiance spectra, which are then compared to those measured

by IASI. In order to fit IASI observations and to minimize the spectral residuals, the method iteratively adjusts the radiative





transfer inputs, namely the dust profile and surface temperature, using Tikhonov–Philips-type regularization, until reaching good agreement for different atmospheric and surface conditions. The a priori dust profile used in every pixel (the same profile for all pixels and all seasons) is a first guess of dust vertical distribution obtained from an average of dust extinction vertical profiles over the Sahara Desert, retrieved from CALIPSO/CALIOP satellite observations (Winker et al., 2009). Once IASI

spectra are fitted, a series of quality checks is performed to screen out cloudy measurements and aberrant retrievals, even though sub-visible cirrus clouds (with AOD below $\sim 0.02$) may be difficult to screen out. Then, the final outputs of AEROIASI are calculated for each unscreened pixel, providing a vertical profile of dust extinction coefficient at 10 µm and the associated DOD by vertical integration of the extinction profile. Using thermal infrared measurements, AEROIASI retrievals are mostly sensitive to coarse aerosols (with a radius roughly greater than $\sim 2$ µm). In fact, the contribution of fine dust particles to total

DOD at 10 µm is expected to be less than $\sim 10$ % (Pierangelo et al., 2005); consequently, the AEROIASI product considered here is the coarse mode DOD at 10 µm. The AEROIASI retrieval offers different sensitivities over land and the ocean. Normally, there is more sensitivity over land, as the surface temperature deviates more from that of the atmosphere above as compared to the case over the ocean. However, the surface emissivity over land is less well known and might induce local biases. Moreover, comparisons conducted between AEROIASI and AERONET coarse AOD retrievals showed distinct discrepancies between

the two datasets in many sites over and downwind of the Sahara Desert (Cuesta et al., 2020) and AEROIASI overestimations far away from desert dust sources (Cuesta et al., 2015). The biases in both cases reach or even exceed 0.1 in absolute value. Additionally, the use of non-zero a priori values for dust abundance (equivalent to an AOD at 10 µm of $\sim 0.03$) is expected to induce positive biases in situations with both very low dust abundances and low sensitivities, as encountered for the relatively lower surface temperatures of mid-latitudes as compared to those near the tropics. Developments for future versions of the

product will aim at screening out these low sensitivity situations.

In this study, we used coarse DOD over the period 2008–2016, provided by the AEROIASI Version 3 dataset, which was retrieved from MetOp-A/IASI data (IASI-A, Level 3), whose mission was completed in November 2021. The horizontal resolution of the AEROIASI dataset is $1° \times 1°$. The DOD at 10 µm was obtained by vertically integrating the extinction coefficient and then it was spectrally converted from 10 µm to 550 nm using a conversion factor of 1.70, derived with a Mie code. The

derived AEROIASI coarse DOD (at 550 nm) considers coarse dust particles larger than 0.6 µm in radius. The spatial collocation between the two datasets was achieved by re-gridding MONARCH coarse DOD through bilinear interpolation using the coarser AEROIASI grid as a reference. Finally, MONARCH was linearly interpolated in terms of time over the exact date-time of the IASI retrievals, as provided by the AEROIASI dataset.

## 2.5 AERONET dust-filtered products

High-quality aerosol optical properties are provided by the ground-based photometer network of AEronet RObotic NETwork (AERONET; Holben et al., 1998; O'Neill et al., 2003; Giles et al., 2019). These instruments rely on extinction measurements of the direct and scattered solar radiation at several nominal wavelengths (between 340 and 1020 nm). In addition, direct-sun AOD processing includes the Spectral Deconvolution Algorithm (SDA) described in O'Neill et al. (2003). This algorithm yields submicron (fine) and super-micron (coarse) AOD at a standard wavelength of 500 nm from which the fraction of fine





mode to total AOD can be computed. The algorithm fundamentally depends on the assumption that the coarse mode Ångström exponent and its derivative are close to zero. AERONET provides a long-term and continuous database of aerosol optical, microphysical and radiative properties, the best currently available on a global basis for aerosol research and characterization, validation of satellite retrievals and evaluation of aerosol models.

The descriptions of the MIDAS, MISR, and AEROIASI dust products above summarized the features and the uncertainties of
the total and coarse DOD products which depend upon the instruments' capabilities, the limitations of the retrieval techniques, and the validity of the assumptions made in order to separate mineral dust aerosols from other aerosol components. All four observational datasets (including AERONET) have their advantages and disadvantages, thus can be complementary to each other in order to overcome limitations regarding the quality of the dust retrievals and the spatiotemporal coverage: MIDAS provides total and coarse DOD observations both over land and sea with the finest spatial resolution ($0.1° \times 0.1°$); MISR
provides the most physically robust separation of DOD by discriminating dust aerosols based on actual retrieved particle shape information; AEROIASI has the most frequent sampling, covering the Earth twice a day, and it is the only dataset to provide nighttime measurements; AERONET ground-based measurements provide the finest temporal resolution ($\sim 15$ minutes), giving the possibility to assess MONARCH at its original 3-hourly time-scale. Moreover, the signal-to-noise ratio for the AERONET direct-sun measurements is high and the surrounding surface reflectance usually makes no significant
contribution to the signal in most cases. This renders AERONET AOD the best available source for surface-based particle property retrieval results, therefore in this study the AERONET dust-filtered retrievals were used not only to assess the model outputs but also to validate the quality of the satellite-based dust products (see Sect. 3).

In this study, we used AERONET Version 3 quality-assured data (i.e., Level 2.0) as a reference dataset (Giles et al., 2019). Since AOD includes contributions from different types of particles, a dust-filter method was applied to identify AOD obser-
vations in which dust is the dominant aerosol type. AERONET dust-filtered AOD (i.e., DOD) is based on direct-sun AOD retrievals between 440 and 870 nm. Although direct-sun does not yield AOD at 550 nm, this variable is calculated from the AOD at 440, 675 and 870 nm and the Ångström Exponent at 440–870 nm (AE) using the Ångström's law. Then AE is used as a filter because it is inversely related to the average aerosol size. Lower AE values ($< 1$) indicate significant presence of coarse-mode particles (e.g., mineral dust and sea-salt), whereas higher AE ($> 1$) values imply a large abundance of fine parti-
cles (e.g., biomass burning and urban aerosols; Papagiannopoulos et al., 2018). Here we follow the discrimination method of Basart et al. (2009), where DOD = AOD when AE $< 0.75$, and all data with AE $> 1.2$ are considered free of dust, i.e., DOD = 0. These two definitions can introduce uncertainties, and in particular, a potential over- and underestimation of the total dust contribution, respectively. Other studies have used lower discrimination thresholds (e.g., AE $< 0.6$), in an effort to obtain pure mineral dust conditions (e.g., Di Tomaso et al., 2022), but thereby excluding more AOD observations in long-range transport
regions. Finally, in this study a mixed aerosol type is assumed when $0.75 \leq$ AE $\leq 1.2$ and the corresponding AOD values are not considered for our analysis.

Regarding AERONET coarse AOD, it was retrieved based on the SDA which yields fine and coarse mode AOD at 500 nm, assuming particle radius 0.6 μm as the inflection point in the volume size distribution. The coarse mode AOD is dominated by maritime/oceanic aerosols and desert dust, whereas other natural sources, such as wildfires, can also produce coarse-mode





aerosols. Sea-salt is usually associated with low AOD ($< 0.03$; Dubovik et al., 2002) and mainly affects coastal stations, and therefore inland high coarse AOD values is assumed to be mineral dust. Moreover, any disparity between the wavelength difference 550 nm and 500 nm is negligible, as coarse mineral dust is wavelength-independent in the visible range (Eck et al., 1999). Therefore, coarse AOD from AERONET SDA will be used as the corresponding AERONET coarse DOD.

Both AERONET dust-filtered retrievals (total and coarse DOD) are dominated by mineral dust; however, small-size particles
(anthropogenic aerosols, biomass burning, etc.) are always present, especially far away from the sources, whereas sea-salt particles can contaminate our retrievals mainly at AERONET stations close to the coast (Basart et al., 2009). Moreover, AERONET particle properties retrieved from sky-scan measurements (e.g., coarse AOD), can be contaminated by the reflectance of the various surface types, such as snow, ice or even some desert surfaces (Sinyuk et al., 2007). Consequently, an overestimate of the AERONET total and coarse DOD is expected.

All the AERONET stations located within the MONARCH reanalysis geographical domain and operating during the reanalysis period were considered, excluding the stations that are at high altitudes ($> 2$ km above sea level). In total, retrievals from 238 stations were used for the present analysis. Total and coarse DOD datasets from the dust regional reanalysis were bilinearly interpolated over each AERONET station. AERONET data are acquired at 15-minute intervals on average; therefore, all AERONET measurements within $\pm 90$ minutes of the model outputs have been averaged for the comparison on a 3-hourly
basis. Figure 2 shows the location of the AERONET sites with at least 30 temporally collocated pairs available.

## 2.6 Evaluation strategy

The evaluation metrics that were used to quantify the level of agreement between the model simulations and the observations are the mean bias (MB), the root mean square error (RMSE), the fractional gross error (FGE) and the correlation coefficient (CC), the definitions of which are given in Appendix A.

The inter-comparison of total and coarse DOD was conducted over two different temporal scales (annual and seasonal) and over two different spatial scales (grid-point and regional). All the statistical indicators (Table A1) were computed on an annual scale, considering all the different model and satellite datasets collocated pairs for the period 2007–2016 of the reanalysis, and on a seasonal scale where the collocated data of a certain season were compared throughout the years according to the following classification: boreal winter (December-January-February: DJF), boreal spring (March-April-May: MAM), boreal
summer (June-July-August: JJA) and boreal autumn (September-October-November: SON). The seasonal subdivision of the datasets allows for the assessment of the MONARCH reanalysis performance in reproducing the annual cycle and the seasonal patterns of the total and coarse DOD. The aforementioned temporal aggregations were generated for each grid-point of the reanalysis-satellite collocated data and for each individual AERONET station.

Moreover, the evaluation statistics were produced at a regional scale in order to assess model performance over regions
with distinct characteristics. The study geographical domain has been divided in ten specific sub-regions (Fig. 1) where the model scores were computed considering all the modelled and satellite-based dust product pairs contained in each one of them, giving the opportunity to identify any dependencies between the different model and satellite datasets and the features of each region. The ten sub-regions are mainly classified into three groups: (i) continental regions that contain the mineral dust





sources, where high DOD is observed throughout the year: Northern Africa, Middle East and Western Asia (hereafter NorAfr,
MidEas and WesAsi, respectively); (ii) remote regions of rare dust events, suitable for reanalysis evaluation under conditions
of very low DOD: North Atlantic, Northern Europe and Russia (hereafter NorAtl, NorEur and Russia); (iii) maritime and
continental regions located downwind of the dust sources, which contain the main dust transport pathways: Tropical Atlantic,
Mediterranean Sea, Arabian Sea and Sub-Saharan Africa (hereafter TroAtl, MedSea, AraSea and SubSah). The latter are
subject to seasonal DOD variation. Furthermore, the borders between the regions are defined so that every region consists
mainly of one surface type (i.e., land or sea). This rough approximation can improve the interpretation of the regional results,
considering that the surface type is associated with the retrieval algorithms used to derive AOD from the satellite observations.

Nevertheless, differences between the inter-comparison results obtained from the different datasets do exist in some cases.
These discrepancies may be due to several possible reasons related to the features of the datasets and the region. The uncer-
tainties involved in the derivation of MIDAS, MISR, AEROIASI, and AERONET dust products inevitably contribute to their
differences too. All satellite-based instruments have increased difficulty retrieving particle properties at low AOD - let alone
the DOD fraction in regions where it is even lower - especially over some surface types for which the reflectance can negatively
impact the retrieval quality. AERONET's discrimination method can also allow large sea salt particles to be misclassified as
dust, especially at coastal sites. Moreover, MISR has much less frequent sampling compared to MIDAS, whereas AERONET's
fine temporal resolution permits the detection of sub-daily micro- and mesoscale dust activity caused by local sources, that
satellites' less frequent sampling can miss. Lastly, unfavorable observing conditions, such as cloud cover common at high lat-
itudes especially during wintertime, in addition to basic sampling frequency, can also decrease the quality of DOD retrievals,
for example due to unmasked cirrus clouds misclassified as dust.

Finally, a multi-sensor aggregation comparison based on the considered satellite-based dust datasets (i.e., MIDAS, MISR and
AEROIASI) is applied to get an overall assessment of the MONARCH performance. We excluded AERONET from the multi-
sensor aggregation at a regional scale because the representativeness of the computed regional metrics remains questionable
due to the uneven distribution of stations in the various sub-regions both quantitatively and spatially (Fig.2). For example, a
large number of network sites sufficiently covers the Mediterranean region and Northern Europe, whereas only one station
corresponds to the Arabian Sea, which is additionally located at the edge of the sub-region.

The regional evaluation metrics of each satellite dataset were averaged to one final value weighted by the number of the
model and satellite-based dust product pairs that each dataset contributes within each sub-region. Even though we consider
the contributions of all the available collocated pairs, it is noted that the different sampling frequencies (temporal resolution)
and overpass times for a given location of the satellites considered in the study, complement each other, providing together
higher temporal coverage. The weighted mean of the statistical indicators was computed at annual and seasonal scale for every
sub-region according to the equations shown in Table A2.





## 3   Satellite-derived dust products inter-comparison with AERONET


A robust model assessment requires the observational data to be reliable and consistent across the study spatial domain, regardless of surface type and the intensity of dust activity. The uncertainties that satellite data can present under certain conditions, as described earlier in Sect. 2, are likely to skew the results of the model assessment. In order to identify the main performance skill of the satellite-derived dust data, in this section we perform a quality check based on comparisons with ground-based

AERONET observations. AERONET data have already been used as a "gold standard" for validating most satellite AOD products. Although MIDAS, MIRS and AEROIASI have been evaluated using dust-related AERONET retrievals in independent analysis (Gkikas et al., 2021; Kahn and Gaitley, 2015; Kalashnikova and Kahn, 2006; Cuesta et al., 2015, 2020), here we seek to assess the performance of the different satellite-based dust products in a common framework (spatial and temporal) for the later comparison with the dust reanalysis.

The comparison between satellite-based dust products and AERONET is performed for each station individually using collocated satellite and ground-based measurements. Each satellite dataset was spatially averaged over the AERONET sites, and the AERONET time-series were temporally averaged centered on the satellite overpass time at the site. The criteria of spatiotemporal coincidence are ±2 hours for AERONET and ±1° of latitude and longitude for AEROIASI, ±0.5° for MISR and ±0.2° for MIDAS, according to the spatial resolution of each satellite dataset used. The time-series that emerged from the

collocation were then compared to each other using the metrics defined in Table A1. In addition, DOD time-series retrieved from MIDAS and MISR (hereafter MIDAS+MISR), as well as coarse DOD retrievals from MIDAS and AEROIASI (hereafter MIDAS+IASI), were combined at station level and then compared to AERONET with the aim to investigate if an aggregated satellite multi-sensor product could statistically mitigate the weaknesses of each sensor and the biased values they introduce into the individual products.

Figure 2 shows the DOD comparison of the satellite MIDAS (Fig. 2, 1st column), MISR (Fig. 2, 2nd column) and MIDAS+MISR (Fig. 2, 3rd column) dust products with AERONET observations. As it is expected, overall all annual DOD values (Fig. 2a–f) shows a marked south-to-north gradient with DODs maxima (above 0.36) in the Sahel (Fig. 1, "D") and the Middle East (latitudes < 30° N) and DOD minima in continental Europe and Russia (under 0.05). The CC between MIDAS and MISR against AERONET (Fig. 2p–q) is very high at all stations affected by dust regularly (CC > 0.8), whereas it drops below 0.4

at sites where the presence of dust is less frequent, reaching even negative values, down to −0.4, at few coastal stations of Northern Europe.

The DOD comparison of MIDAS and MISR against AERONET shows underestimations (MB < 0) at most sites situated close to or around the dust sources in Northern Africa and the Middle East and slight overestimations in Europe (MB up to 0.04). In particular, MIDAS largest underestimations (MB < −0.1) are recorded at stations located along with the dust outflow

from the Sahara Desert to the Gulf of Guinea and the Atlantic Ocean, in agreement with Gkikas et al. (2021) and Wei et al. (2019), as well as at some stations in the Arabian Peninsula and Western Asia on the coastline with the Arabian Sea (Fig. 2g). At stations located in the western Sahara Desert and the western part of the Sahelian Belt MIDAS shows low RMSE (up to





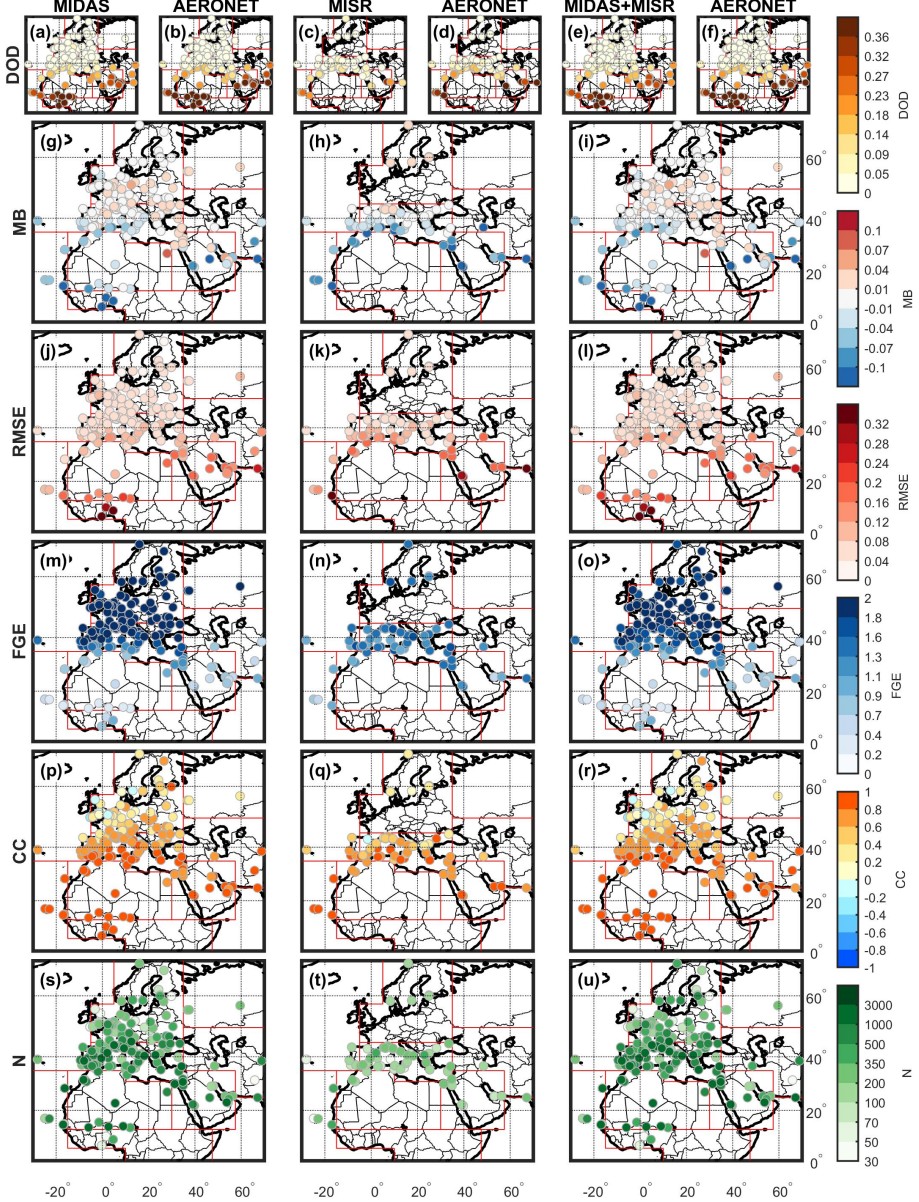

**Figure 2.** DOD comparison of MIDAS (1st column), MISR (2nd column) and MIDAS+MISR (3rd column) against AERONET for the period 2007–2016. The metrics MB, RMSE, FGE and CC (Table A1) were computed at station level. The obtained scores are presented here only for sites with N ≥ 30 collocated pairs. The red frames in the background delimit the sub-regions defined in Fig. 1.

0.12; Fig. 2j) and FGE (up to 0.7; Fig. 2m) considering the high DOD values (annual DOD mean above 0.36 for both MIDAS and AERONET).





Similarly, MISR presents overall underestimation (MB $< -0.07$), particularly at all the stations along the coastlines of
Africa and the Middle East (MB $< -0.1$) (Fig. 2h). In the Mediterranean Sea and continental Europe, MISR shows smaller
differences against AERONET associated with the lower DODs in these long-range transport regions (annual DOD mean up
to 0.09 for MISR and AERONET) with RMSE up to 0.08 (Fig. 2k), MB ranges between $-0.04$ and 0.04 and FGE achieving
maximum values (up to 2; Fig. 2n).

Lastly, the combination of MIDAS and MISR DOD at station level into a single time series (i.e., MIDAS+MISR) compared
to AERONET do not show major deviations with respect to the independent MIDAS and MISR datasets, and at most sites
they are identical to MIDAS scores. This is due to the fact that MISR contributes to less than 40 % of the 238 available sites
and that is only at coastal stations. Moreover, the impact of each sensor to the final product is determined by the number
of measurements available at each station (Fig. 2s–t), where in most sites MIDAS exceeds MISR sampling in number N of

observations, due to its higher temporal resolution.

    The coarse DOD comparison with MIDAS and AEROIASI against AERONET (Fig. 3) shows different results. As expected,
overall annual coarse DOD values (Fig. 3a–f) show a marked south-to-north gradient with DODs maxima in the Arabian
Peninsula ($> 0.23$ for both sensors) and in the Sahel ($> 0.27$ for MIDAS and no more than 0.23 for AEROIASI) and DOD
minima in continental Europe and Russia ($< 0.05$ for MIDAS but no less than 0.09 for AEROIASI). The MIDAS CC map shows

a very clear correlation (CC $> 0.8$) with AERONET coarse DOD over all the dust source regions and the Mediterranean Sea,
and a fairly high correlation (CC $> 0.6$) at most sites in Northern Europe (Fig. 3p). On the other hand, AEROIASI CC ranges
between 0.4 and 0.8 at AERONET sites located up to 40° N, whereas no correlation (CC $\sim 0$) or even negative correlation was
computed at all sites across Northern Europe and Russia (Fig. 3q), showing significant weakness in reproducing the temporal
evolution of coarse DOD in those regions. Similar tendencies are found for RMSE (Fig. 3j–k) and FGE (Fig. 3m–n) between

the two satellite-derived dust products; however, AEROIASI provides relatively greater errors compared to MIDAS, at almost
all AERONET stations, and for both metrics, something that affects the multi-sensor product as well, especially in northern
latitudes (Fig. 3l and o).

    Overall, MIDAS underestimates the coarse DOD compared to AERONET (MB ranges from 0.01 in Europe to less than $-0.1$
in the Sahel; Fig. 3g) whereas AEROIASI shows overestimations (MB $> 0.04$) almost everywhere except for the Sahel, Gulf

of Guinea, Capo Verde and the Persian Gulf (MB $< -0.04$; Fig. 3h). The results of MIDAS coarse DOD in Europe (with MB
$< 0$; Fig. 3g) with respect to MIDAS total DOD results (with MB $> 0$; Fig. 2g) emphasizes the fact that the size distribution of
MIDAS is skewed toward finer fractions. This is directly connected with the use of the MERRA-2 reanalysis fine/coarse DOD
ratio for the MIDAS total and coarse DOD estimations (see Sect. 2.2). As pointed out by Buchard et al. (2017), MERRA-2
shows a larger contribution of dust fine fractions to the total dust budget. Regarding AEROIASI, MB results (Fig. 3h) are

consistent with the findings of previous studies, namely MB ranges from $-0.1$ to 0.1 over the Sahara Desert (Cuesta et al.,
2020) and overestimation of coarse DOD reaches 0.1 far from the desert dust sources (Cuesta et al., 2015). Positive biases
encountered north of 40° N are most likely linked to the use of non-zero a priori values for the retrieval. When the abundance
of dust and the product sensitivity too are low (as frequently expected north of 40° N), the Tikhonov–Philips inversion used by

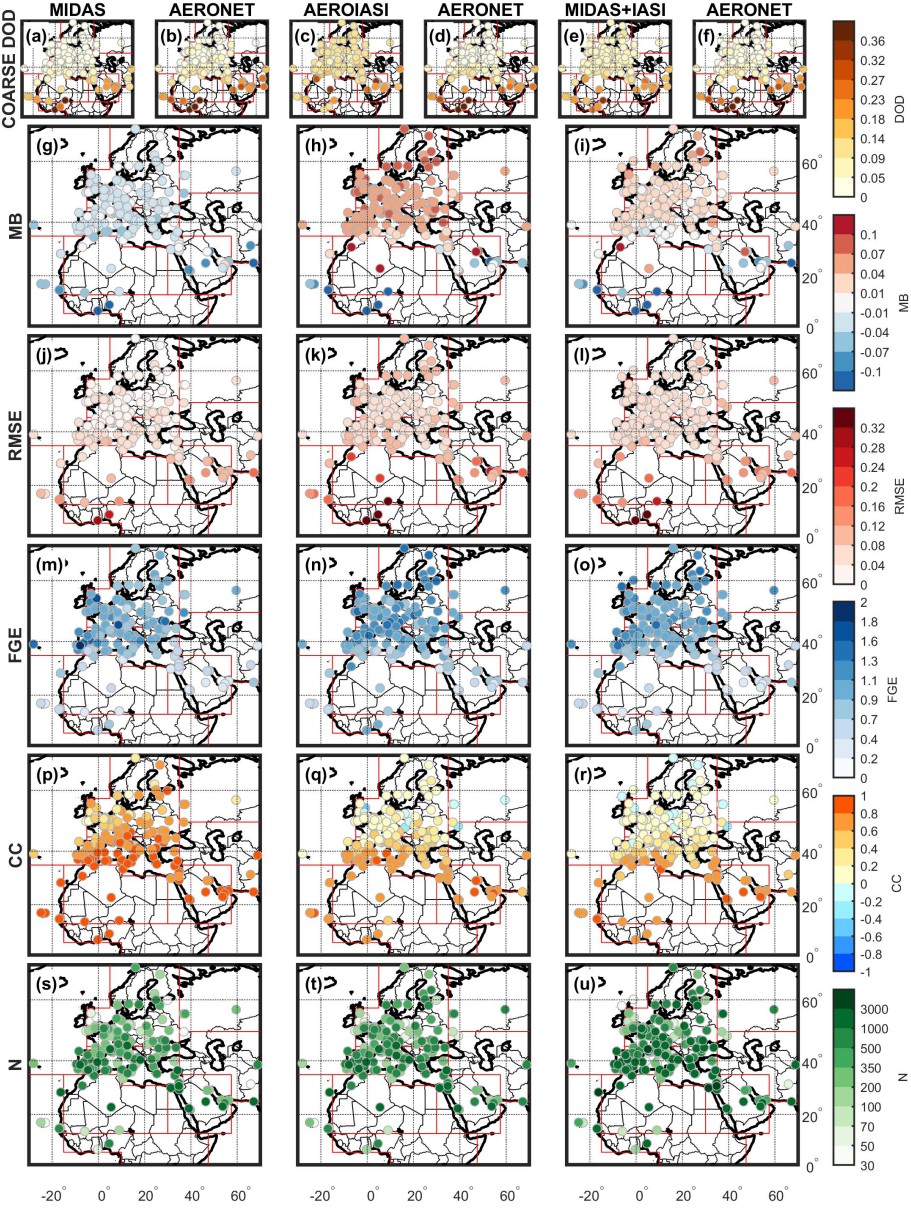

**Figure 3.** Coarse DOD comparison of MIDAS (1st column), AEROIASI (2nd column) and MIDAS+IASI (3rd column) against AERONET for the period 2007–2016. The metrics MB, RMSE, FGE and CC (Table A1) were computed at station level. The obtained scores are presented here only for sites with N ≥ 30 collocated pairs. The red frames in the background delimit the sub-regions defined in Fig. 1.

AEROIASI tends to provide the a priori value which are clearly visible in terms of long-term averages (as in the case of Fig.
440 3).





The MIDAS+IASI product was derived from the aggregation of the two datasets (Fig. 3e), to which they contribute equally at stations located at lower latitudes, whereas AEROIASI's impact is bigger at sites in Northern Europe and Russia, owing to the higher number of IASI measurements available in those regions (Fig. 3s–u). MIDAS+IASI CC (Fig. 3r) provides low correlation at all AERONET stations in Northern Europe (latitudes > 45° N). MIDAS+IASI shows strong underestimation (MB

< −0.1) in the southwest of the Sahara Desert and on the coast of Pakistan, whereas overestimation is observed in Morocco (MB > 0.1) and Europe (MB up to 0.07 in northern latitudes; Fig. 3i). These overestimates in Europe are directly associated with the strong overestimations of AEROIASI (Fig. 3h). MIDAS+IASI is in good agreement with coarse DOD AERONET along the northern coast of Africa and the Red Sea, and in most sites across the Mediterranean Sea (−0.01 < MB < 0.01). As a result, the MIDAS+IASI coarse DOD product is more reliable over dust-rich regions and becomes more uncertain in regions

of sporadic dust events, although, overall its performance is poorer than the only-MIDAS' coarse DOD.

## 4 MONARCH reanalysis assessment

In this section the assessment of total DOD and coarse DOD products of the MONARCH reanalysis for 2007–2016 is analyzed. Firstly, MONARCH is compared versus each observational-based dust dataset (i.e., AERONET, MIDAS, MISR and AEROIASI) at station level in the case of AERONET and at grid-cell level considering the individual grids for each satellite

dataset. Then, the comparison was made at regional scale by generalizing the results based on the ten sub-regions shown in Fig. 1. The regional scores were computed at two different temporal scales as well: annual and seasonal. Finally, an overall assessment is attempted through the aggregation of the regional results that were obtained by the evaluation against the satellite datasets.

### 4.1 Independent dataset analysis

Starting with the MONARCH DOD assessment, Fig. 4 shows the results of the comparison with DOD products retrieved from the space-based and ground-based observations. At first glance, MONARCH seems to capture the DOD spatial distribution obtained by the all three observational datasets (i.e., MIDAS, MISR and AERONET), reproducing the major dust hotspots and the dust transport pathways in the area (Fig. 4a, c and e). More specifically, the reanalysis DOD exceeds 0.27 over all the dust sources listed in Fig. 1, with values exceeding 0.36 over the western Sahara Desert, the Bodélé Depression (Fig. 1, "E"), the

Sahel and the Arabian Peninsula (Fig. 4a and e). Moreover, a pronounced dust plume is simulated stretching across the Tropical Atlantic Ocean. The magnitude and latitudinal extent are greatest over the west African coastline with the maximum DOD up to 0.32 and gradually decreasing westward towards the central Tropical Atlantic, as expected for a dust plume that originates in Africa. Similarly, moderate dust transport is simulated over the adjacent regions of the Mediterranean and the Arabian Sea with maximum DOD values up to 0.18 and 0.23 respectively, closer to the dust sources (Fig. 4a and c).

The comparison between MONARCH and MIDAS shows a strong correlation over the entire domain, with CC maxima (> 0.8) found throughout the Sahara Desert, the Sahel Belt, the Middle East and the Tropical Atlantic and partially over the Arabian Sea, the Mediterranean Sea and even in the North Atlantic (Fig. 4p). The correlation between reanalysis and MISR DOD (Fig.





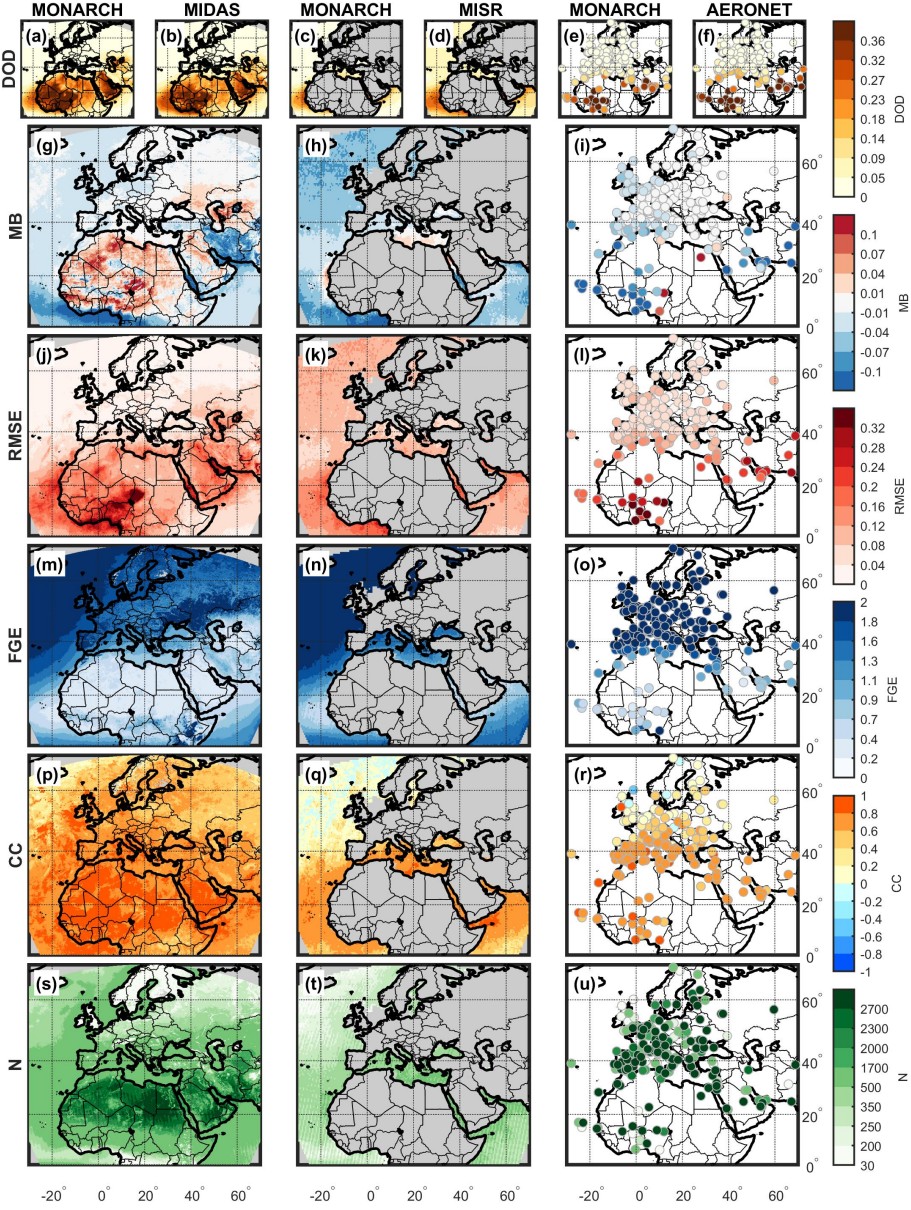

**Figure 4.** Spatial distribution of the total DOD simulated by the MONARCH reanalysis (a, c and e) and collocated with observations by MIDAS (b), MISR (d) and AERONET (f), along with the respective statistic parameters: MB (g–i), RMSE (j–l), FGE (m–o) and CC (p–r). Number N gives the total number of pairs collocated during the study period 2007–2016, at grid (s and t) and station level (u).

4q) is higher (CC > 0.6) around the dust source areas but is poorer over the North Atlantic, reaching zero or negative values, presumably associated with MISR's limited sampling compared to MIDAS (Fig. 4s–t). The reanalysis is highly correlated with
AERONET observations over sites affected by medium-range dust transport, whereas the CC diminishes (< 0.4) towards the





northern latitudes of the study region, especially close to coastal areas (Fig. 4r), where the number of observations used for the comparison is $< 250$ (Fig. 4u). Furthermore, RMSE (Fig. 4j–l) and FGE (Fig. 4m–o) spatial distributions are similar in the comparison of the reanalysis with MIDAS, MISR and AERONET, showing maximum RMSE values ($> 0.24$) and minimum FGE values ($\sim 0$) in the regions more affected by the presence of mineral dust with high DOD ($> 0.18$ on annual average),

as well as minimum RMSE values ($\sim 0$) and maximum FGE values ($\sim 2$) in the long-range transport regions (annual DOD $< 0.05$).

Overall, the MONARCH reanalysis tends to underestimate DOD except in desert dust source regions where MONARCH and the observational datasets show some discrepancies. In the comparison with MIDAS (Fig. 4g), the reanalysis shows overall overestimations in Northern Africa, the Arabian Peninsula and the part of Kazajistan, Uzbekistan and Turkmenistan with larger

overestimation (MB $> 0.1$) in the dust sources of Algeria, over the Sahel, the Bodélé Depression and the Ustyurt Plateau (Fig. 1, "A", "B", "D", "E" and "J") and underestimation (MB $< 0$) in the Persian Gulf and the arid regions of Iran and Afghanistan (Fig. 1, "K"). Regarding the comparison with AERONET (Fig. 4i), MONARCH presents an overall underestimation, with MB $< -0.1$ in western part of the Sahel and near to the coast of the Gulf of Guinea, except in the downwind sites of the Bodélé Depression and over the Great Sand Sea (Fig. 1, "F") where maximum overestimations (MB $> 0.1$) are observed. Over

continental Europe and Russia, near-zero MB is observed compared to both datasets, because of the relatively lower DOD by MIDAS and AERONET and simulated by MONARCH in those regions. Over the maritime regions, the comparison with MIDAS, MISR and AERONET shows similar results. The reanalysis strongly underestimates the dust transport towards the Gulf of Guinea and in Cape Verde (MB $< -0.1$), whereas it moderately underestimates (down to $-0.07$) and overestimates (up to 0.04) over the Tropical Atlantic and the central and eastern Mediterranean Sea, respectively. Particularly over the North

Atlantic and the Arabian Sea, the comparison of MONARCH with MISR shows higher RMSE values (up to 0.16; Fig. 4k) and larger underestimations ($-0.04 <$ MB $< -0.07$; Fig. 4h) compared to MIDAS ($-0.01 <$ MB $< -0.04$; Fig. 4g), because of the relatively higher DOD recorded by MISR in these regions (Fig. 4d). This difference between MISR and MIDAS DOD can be traced to the difference between MISR and MODIS total AOD, as in previous studies MISR AOD was found to be generally larger than MODIS AOD over water (Guo et al., 2013; Abdou et al., 2005; Kahn et al., 2010).

The comparison of MONARCH DOD with each observational dataset was made at regional level as well, based on the ten sub-regions shown in Fig. 1. The statistical parameters were computed at regional scale and at two different temporal scales, annual and seasonal, and are presented in Supplement. In particular, the regional results of the MONARCH comparison against MIDAS, MISR and AERONET are shown in Fig. S1, Fig. S2 and Fig. S3, respectively. The seasonal and regional DOD patterns show good agreement between MONARCH and MIDAS (Fig. S1), identifying MAM and JJA as the seasons of

maximum dust emissions from the sources (DOD $> 0.3$ in NorAfr and MidEas). In MAM the meteorological conditions favor the transport of dust from the southern parts of the Sahara (e.g., Bodélé Depression) to the Sahel (DOD $\sim 0.3$ in SubSah), and from the northern Sahara sources and the Syrian Desert (Fig. 1, "G") towards the Mediterranean. In JJA dust plumes are directed from the Sahara and the Middle East towards the Atlantic and the Arabian Sea, respectively. This dust transport seasonality is also confirmed by the seasonal values of MONARCH and MISR over the maritime regions (Fig. S2), which

are fully covered by MISR dark-water retrievals. On the other hand, the low sensitivity of MISR non-spherical AOD over the





remote regions (NorAtl, NorEur and Russia) is evident here, leading to overestimated annual and seasonal MISR DOD values (annual DOD > 0.05), higher biases (annual MB < −0.05 and RMSE > 0.09) and lower CC (< 0.21) with MONARCH. Lastly, the regional means obtained from MONARCH versus AERONET comparison (Fig. S3) should be used with caution because the sub-regions are not evenly represented by AERONET stations in terms of amount and spatial distribution (see Fig.

2, 3$^{rd}$ column). The best coverage is found in MedSea and NorEur, in MidEas there are fewer stations but well distributed, whereas in NorAft the majority of the stations are located at the edges of the Sahara. The DOD seasonality is again identified over the dust emission and transport regions; however, huge biases between the two datasets like those obtained in SubSah during DJF (MB = −0.45, RMSE = 0.62) can be attributed to mesoscale processes like Haboobs (Roberts and Knippertz, 2012) that can affect ground-based measurements (i.e., AERONET), but remain undetected by the model or even by satellites

due to coarser spatiotemporal resolution. Here we should note that over the dust-rich regions the regional AERONET DOD is significantly larger than the corresponding satellite-derived DOD because the method used to retrieve AERONET DOD excludes cases of mixed aerosol type (see Sect. 2.5), which increases the contribution of pure dust events to the sample, and this eventually increases the mean AERONET DOD.

Repeating the same process, the MONARCH coarse DOD is compared against MIDAS, AEROIASI, and AERONET (Fig.

5). As the coarse DOD is a fraction of the total DOD, the annual mean coarse DOD of MONARCH, MIDAS, and AERONET shows the same spatial distribution as total DOD (Fig. 4). The temporal correlation between the MONARCH reanalysis and MIDAS, AEROIASI and AERONET (Fig. 5p–r) is generally higher near source and transport areas (CC up to 0.8), and diminishes towards the northern latitudes (i.e., north of 40° N). In fact, the comparison with AEROIASI even shows a negative correlation at these latitudes (CC < 0, Fig. 5q). RMSE (Fig. 5j–l) and FGE (Fig. 5m–o) spatial distribution is similar among

MIDAS, AEROIASI and AERONET showing maximum RMSE (> 0.32) and minimum FGE (< 0.4) in the regions with the strongest dust activity where the maximum absolute MB was also found (e.g., Bodélé Depression; Fig. 5g–i). In long-range transport regions, AEROIASI presents larger errors (RMSE > 0.08 and MB < −0.07) than MIDAS and AERONET. The MONARCH reanalysis overestimates the coarse DOD over all the dust sources when compared to MIDAS and AEROIASI (Fig. 5g–h) with values that exceed 0.1 over the Bodélé Depression and its downwind areas as well as over the major dust

sources of the western Sahara Desert. The comparison against AERONET shows overall underestimations (MB < 0) with maxima (MB < −0.1) at stations situated downwind of the Bodélé Depression towards the Gulf of Guinea, in Cape Verde and close to the Registan Desert (Fig. 5i). The overestimations of MONARCH in the comparison with MIDAS over desert dust sources (Fig. 5g) are related to the fact that the size distribution of MIDAS is skewed toward finer sizes (see Sect. 3). Moreover, slight underestimations in Europe in the comparison against AERONET (Fig. 5i) can be attributed to the discrimination method

applied (see Sect. 2.5) that can also allow large sea-salt particles or other coarse aerosol of local origin. The reanalysis is in very good agreement with MIDAS over the remote regions of the North Atlantic and continental Europe, where MB is almost zero, whereas the comparison with AEROIASI away from the dust sources produced a very strong underestimate (MB < −0.1). Moreover, the MB in Fig. 5h changes abruptly when moving from desert to remote regions because the coarse DOD provided by AEROIASI is consistently larger than 0.09 over the entire domain, even in remote regions, whereas it does not exceed 0.36

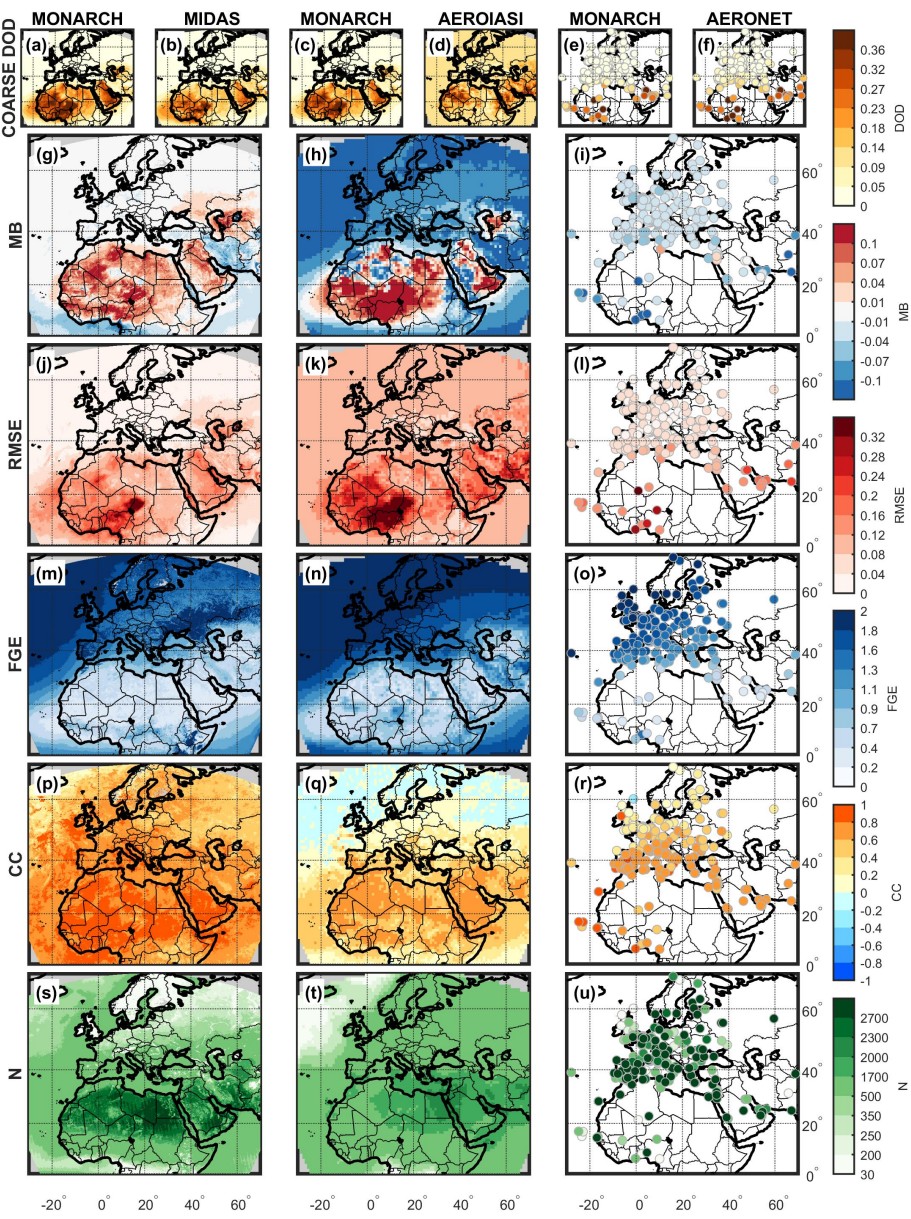

**Figure 5.** Spatial distribution of the coarse DOD simulated by the MONARCH reanalysis (a, c and e) and collocated with observations by MIDAS (b), AEROIASI (d) and AERONET (f), along with the respective statistic parameters: MB (g–i), RMSE (j–l), FGE (m–o) and CC (p–r). Number N gives the total number of pairs collocated during the study period 2007–2016, at grid (s and t) and station level (u).

over the dust sources (Fig. 5d), in agreement with the findings in Sect. 3, that AEROIASI tends to underestimate coarse DOD close to desert dust sources and to overestimate it far away from them (Fig. 3h).





Finally, the regional metrics of the MONARCH coarse DOD, compared to MIDAS, AEROIASI and AERONET, were computed at annual and seasonal scale and are presented in Fig. S5, Fig. S6 and Fig. S7, respectively. Naturally, regional coarse DOD follows the cycle of the total DOD over the sub-regions associated with dust emission (NorAfr, MidEas and

WesAsi), as the seasonal means of MIDAS, AEROIASI and the collocated MONARCH data show (Fig. S5 and Fig. S6). The intra-annual variability of the coarse dust particles long-range transport is well represented by MIDAS and MONARCH seasonal coarse DOD over TroAtl, AraSea and MedSea (maxima in JJA, JJA and MAM, respectively). The two datasets are in good agreement too, providing very low seasonal and annual MB values in those sub-regions. On the other hand, AEROIASI does not exhibit any seasonality over the maritime and the remote regions located north of 40° N, with no season-on-season

changes in coarse DOD, which remain consistently greater than 0.1 (NorAtl, NorEur and Russia), probably linked to lack of sensitivity. AEROIASI shows a weak performance for detecting low DOD values in seasons and in regions where minimal dust activity is expected, as already shown in Fig. 3h. As a consequence, large biases (MB $< -0.08$ and RMSE $> 0.09$) and no correlation (CC $\sim 0$) are obtained by the comparison with MONARCH coarse DOD. Regarding the MONARCH versus AERONET comparison at regional and seasonal scale (Fig. S7), the typical patterns in seasonality are also found here. The

seasonal MB and RMSE values over the dust emission and dust transport regions correspond to those obtained by the total DOD comparison. On the other hand, over the remote regions, MB is higher in absolute terms because AERONET tends to overestimate coarse DOD ($> 0.03$) in northern latitudes ($> 40°$ N), providing greater seasonal and annual values than for total DOD (Fig. S3). Nevertheless, the normalized metrics (i.e., FGE and CC) are overall better for coarse DOD than for total DOD (Fig. S3).

## 4.2   Assessment through multi-sensor aggregation

Overall, the DOD comparison results against the three independent datasets (i.e., MIDAS, MISR and AERONET) shows that over the areas of most interest, with high dust activity throughout the year and greater number of measurements, the evaluation scores are consistent, despite the different features of each dataset, namely the raw data, the dust separation assumptions, the uncertainties and the spatiotemporal resolution and coverage. Consequently, the validity of the evaluation results is enhanced,

leading to safer conclusions about the performance of the model. Nevertheless, differences between the results obtained from the different datasets do exist in some cases. In particular, the results obtained by MIDAS and MISR over the North Atlantic differ in MB, RMSE and CC, and the same applies to CC between MIDAS and AERONET.

As shown in Fig. 4, in most regions that are subject to high DOD levels and frequent dust intrusions, the assessment results are not affected by the features of each dataset; instead, datasets from different sensors can be used in a complementary way

to provide more solid insight into the performance of the MONARCH reanalysis. In particular, the regional results obtained from MIDAS (Fig. S1) and MISR (Fig. S2) were averaged to a final weighted mean (MIDAS+MISR), considering as weight the corresponding number N as it is described in Sect. 2.6, using the equations shown in Table A2. As mentioned earlier, the two datasets not only do not overlap but complement each other in terms of space and time, favoring the combination of their results. In particular, MIDAS covers Aqua's overpass time (13:30 LT), both over land and sea whereas MISR considers Terra's

overpass time (10:30 LT) over sea. Moreover, MISR and MIDAS have different spatial and temporal resolution. Consequently,





the two reanalysis samples obtained by collocation with the satellite datasets are complementary to each other. The combined regional results from MIDAS and MISR are presented by region and by temporal scale, in Fig. S4; furthermore, the annual scores are displayed in maps in Fig. 6, to better understand the geographical distribution of the results.

The MONARCH DOD (Fig. 6a) and the multi-sensor aggregation DOD (Fig. 6b) present similar patterns, with small dis-
crepancies in DOD values over and around the dust sources. In most sub-regions MB (Fig. 6c) ranges between $-0.02$ and $0.02$, which is quite low in most cases compared to DOD values. Minimum differences are found in Russia (MB $\sim 0$) as well as in dust source and outflow regions like SubSah (MB $= 0.01$), MidEas (MB $= -0.01$) and MedSea (MB $\sim 0$). The largest positive difference is found in NorAfr (MB $= 0.04$) which contains the Sahara Desert and the highest annual regional DOD values are simulated or observed there (DOD $= 0.29$ and $0.25$ respectively); whereas, the lowest negative MB ($-0.04$) is found in TroAtl
and in AraSea, which are subject to frequent dust transport from the Sahara Desert and the Arabian Peninsula, respectively.

The regional RMSE (Fig. 6d) shows that the greatest differences between reanalysis and observations occur in regions with the strongest dust activity, with values higher than $0.15$ in MidEas, NorAfr, TroAtl and SubSah, whereas the minimum RMSE ($0.03$) occurred in the remote region of NorEur. Conversely, the regional FGE (Fig. 6e) is maximized ($> 1.8$) over the remote regions of NorAtl, NorEur and Russia, whereas the lowest values (FGE $< 0.5$) are found in MidEas and NorAfr. The regional
CC (Fig. 6f) shows a clear north to south positive gradient with minimum value (CC $= 0.49$) over Russia and maximum values (CC $> 0.8$) in NorAfr and SubSah.

Considering all the results obtained from the evaluation metrics at annual scale, as listed in Fig. S4 and visualized in Fig. 6, among sub-regions where high dust concentrations are often observed by satellites (MIDAS+MISR DOD $> 0.07$) the best combined scores are found in MidEas (MB $= -0.01$, FGE $= 0.43$), NorAfr (FGE $= 0.43$, CC $= 0.84$) and MedSea (MB
$= 0$, RMSE $= 0.09$), demonstrating very good reanalysis performance in reproducing the DOD levels and its spatiotemporal variability over the major dust source regions of MidEas and NorAfr as well as in the nearby outflow region of MedSea. Considering that in NorAfr and MidEas the two highest mean annual DOD values were recorded (MIDAS+MISR DOD $= 0.25$ and $0.24$, respectively), the lowest FGE implies that the biases with respect to the MONARCH reanalysis are insignificant there. Moreover, the highest CC ($0.84$) in NorAfr indicates the model's correct simulation of the time evolution of dust emissions
from the Sahara Desert. Similarly, in MedSea MONARCH MB reaches the perfect score, with quite low deviations from it according to RMSE, implying a very good performance of the reanalysis under low DOD conditions (MIDAS+MISR DOD $= 0.07$) and sporadic dust intrusions throughout the year. Furthermore, the fact that those three sub-regions have the two top largest (NorAfr and MidEas) and the fourth largest (MedSea) N (Fig. 6g), shows the consistency of the reanalysis when evaluated against a large number of observations and corroborates the evaluation results.
On the other hand, the combination of the regional annual scores reveals weak agreement between reanalysis and observa-
tions over TroAtl (MB $= -0.04$, RMSE $= 0.16$) and WesAsi (MB $= -0.02$, CC $= 0.66$). The inter-comparison of the MI-
DAS+MISR and MONARCH DOD against AERONET observations (Fig. 2, 3rd column and Fig. 4 3rd column, respectively) shows that MIDAS+MISR DOD is in better agreement with AERONET compared to the MONARCH reanalysis, providing better scores in most of the common sites included in those two regions. In particular, the reanalysis underestimates AERONET
DOD by at least $0.1$ (MB $< -0.01$) at the majority of the stations located in TroAtl and WesAsi, where MIDAS+MISR biases



**Figure 6.** Regional weighted annual mean of the MONARCH reanalysis DOD (a), the MIDAS+MISR DOD (b), and their MB (c), RMSE (d), FGE (e) and CC (f) along with the total regional N (g). The results refer to the study period 2007–2016.

are usually lower than 0.07 (MB > −0.07). As far as TroAtl is concerned, this indicates that the MONARCH reanalysis simulates higher dust deposition rates underestimating the real amount of dust particles that travels towards the Atlantic Ocean;





however, it captures very well the DOD spatiotemporal variability, as reflected by the strong correlation (CC = 0.8) found in this region (Fig. S4).

Lastly, the ability of the reanalysis to correctly simulate the annual cycle of the regional dust activity was examined by computing the seasonally weighted means of the evaluation metrics for winter (DJF), spring (MAM), summer (JJA) and autumn (SON) (Fig. S4). In general, atmospheric dust emissions and transport are subject to seasonal variation, following the changes in wind conditions at synoptic scale; consequently, each sub-region presents a distinct seasonality. At first glance it can be seen that the color patterns of simulated and observed DOD are very similar, showing that the reanalysis captures very

well the annual dust cycle over all sub-regions. Particularly in regions where the presence of dust is predominant, the reanalysis reproduces the DOD peaks in MAM and JJA; correspondingly, the minimum seasonal DOD values are correctly simulated in DJF and SON. Accordingly, the MB and RMSE seasonal patterns are directly related to the seasonal DOD variation, with maximum values in MAM and in JJA which weaken during SON or DJF, for all the sub-regions of interest. On the contrary, in northern sub-regions, patterns in seasonality are less pronounced because of the low presence of dust. The sign of the seasonal

MB agrees with the sign of the annual MB, with the exception of SubSah, where low underestimations during JJA (MB = −0.02) are likely associated with mesoscale convective dust storms (e.g., Haboobs) that the model cannot simulate.

The reanalysis performance in reproducing the regional dust cycle can be also assessed by the normalized parameters of FGE and CC which are less magnitude-dependent and are expected to remain invariable throughout the year. In fact, low FGE seasonal values exhibit remarkable intra-annual stability over the major dust emission regions of NorAfr ($0.39 \leq$ FGE $\leq 0.49$)

and MidEas ($0.41 \leq$ FGE $\leq 0.47$), where intense dust activity and strong seasonal variability have been recorded. In addition, over the main dust transport region of TroAtl, the seasonal fluctuation of the FGE is insignificant. Likewise, almost all the sub-regions of interest present high CC values and weak seasonality, namely NorAfr ($0.81 \leq$ CC $\leq 0.85$), MidEas ($0.72 \leq$ CC $\leq 0.81$), TroAtl ($0.76 \leq$ CC $\leq 0.82$), AraSea ($0.72 \leq$ CC $\leq 0.77$) and MedSea ($0.74 \leq$ CC $\leq 0.76$). The minimum seasonal CC (0.38) found over Russia in DJF, along with all the scores for the same season computed in Russia and in NorEur, should

be considered unreliable because the number of dust retrievals decreases significantly in the north of Europe during the DJF season (N $\sim 0 \times 10^7$ in Russia and N $= 0.15 \times 10^7$ in NorEur), as MIDAS covers only snow-free surfaces.

Following the same methodology, the coarse DOD comparison results from MIDAS (Fig. S5) and AEROIASI (Fig. S6) were combined on a regional scale. Again here the satellite datasets complement each other since the MODIS equatorial overpass time is 13:30 LT while IASI crosses the equator twice a day at 09:30 and 21:30 LT. The MIDAS+IASI scores are presented by

region in Fig. S8 from which the annual means are illustrated in Fig. 7.

Deviations in coarse DOD between reanalysis (Fig. 7a) and MIDAS+IASI (Fig. 7b) are evident in SubSah (MB = 0.04; Fig. 7c), and mainly in NorAfr where the maximum overestimation is observed (MB = 0.06), which is 50 % greater compared to the DOD MB over the same sub-region (Fig. 6c). This is a significant overestimation if we consider that the coarse DOD is a fraction of the total DOD. On the other hand, quasi-zero MB's were recorded in WesAsi, Russia and MedSea. Accordingly,

the RMSE (Fig. 7d) exceeds 0.12 in NorAfr, SubSah and MidEas, whereas the best scores (RMSE $\leq 0.04$) are recorded in NorAtl, NorEur and Russia. In contrast, the lowest FGE values ($\leq 0.5$) can be found in MidEas and NorAfr and the highest (FGE $\geq 1.9$) over the remote regions of NorAtl, NorEur and Russia (Fig. 7e). Lastly, the CC (Fig. 7f) exceeds 0.8 in NorAfr





**Figure 7.** Regional weighted annual mean of the MONARCH reanalysis coarse DOD (a), the MIDAS+IASI coarse DOD (b), and their MB (c), RMSE (d), FGE (e) and CC (f) along with the total regional N (g). The results refer to the study period 2007–2016.

and SubSah, while over Russia a very low correlation is observed (CC = 0.37), as the negative CC values recorded there during the MONARCH reanalysis versus AEROIASI comparison (Fig. 5q), affect the corresponding regional values (Fig. S6).





Considering all the regional annual results (Fig. S8), among sub-regions where high dust concentrations are often observed by satellites (MIDAS+IASI coarse DOD > 0.05), the best scores are found in AraSea (FGE = 0.74, CC = 0.78), MedSea (MB = 0.01, RMSE = 0.07) and TroAtl (MB = −0.01, CC = 0.79), which contain the main dust outflow pathways with moderate and sporadic coarse dust activity throughout the year. On the other hand, the reanalysis seems to perform less efficiently over the African continent, namely in NorAfr (MB = 0.06, RMSE = 0.14) and in SubSah (MB = 0.04, RMSE = 0.14), where

MONARCH seems to generate a surplus of coarse dust particles. Given that both sub-regions present the highest CC (0.83), the coarse dust emissions from the Sahara Desert seem to be overestimated by a constant factor during the entire study period.

    Focusing in those two regions, the inter-comparison of the MIDAS+IASI and MONARCH coarse DOD against AERONET (Fig. 3, 3$^{rd}$ column and Fig. 5 3$^{rd}$ column, respectively) shows that MONARCH coarse DOD is in better agreement with AERONET compared to MIDAS+IASI, providing better scores in most of the common sites. Especially at common sites

located in the Sahara Desert, MONARCH MB is limited between −0.04 and −0.01 showing very small differences with AERONET (Fig. 5i, see NorAfr) considering that AERONET coarse DOD at these sites exceeds 0.14 (Fig. 5f, NorAfr); whereas MIDAS+IASI MB varies from −0.1 to 0.1, implying significant deviations from the ground-based measurements (Fig. 3i, NorAfr). Actually, the MIDAS+IASI product at those sites is mostly biased due to AEROIASI's strong over- and underestimations (Fig. 3h, NorAfr), whereas MIDAS reproduces quite well AERONET coarse DOD (−0.04 < MB < 0; Fig.

3g, NorAfr). On the other hand, the MONARCH annual regional scores commented in the previous paragraph are dominated by MIDAS contribution (see annual values of Fig. S5 and Fig. S8), due to a larger number of MODIS observations. Considering that MIDAS coarse DOD is derived using 0.5 μm as a cut-off radius whereas MONARCH uses 0.6 μm, it means that in case of common radius value, the MONARCH versus MIDAS MB results over NorAfr and SubSah would be even larger. This again should be attributed to MERRA-2 fine/coarse DOD ratio which eventually underestimates MIDAS coarse DOD, especially

over areas of high dust activity (Fig. 3g). In conclusion, the discrepancies between MONARCH coarse DOD and both satellite datasets over the African continent are most likely due to underestimations in MIDAS and AEROIASI coarse DOD retrievals.

    Any seasonality in the performance of the reanalysis in reproducing the coarse DOD can be assessed by the seasonal values of the metrics computed at regional scale (Fig. S8). The reanalysis simulates very well the annual cycle of coarse dust at both emission and transport regions where coarse DOD seasonality is intense and peak activity occurs in MAM or in JJA, depending

on the sub-region, exactly as was observed by the two satellite instruments. The discrepancies are more pronounced and subject to seasonal variation over the main dust source regions of NorAfr and MidEas, where MB is maximized during JJA, when coarse dust loads are higher, and weakens during SON. A reverse pattern is recorded in SubSah, TroAtl and MedSea where the seasonal minima (MB ∼ 0) corresponds to seasons of quite high coarse DOD values (i.e., JJA, DJF and MAM, respectively). Moreover, in MedSea and mostly in WesAsi seasonal MB remains remarkably low and stable throughout the year, despite the

seasonal coarse DOD fluctuations. On the other hand, RMSE seasonal pattern follows consistently the corresponding coarse DOD annual cycle of each sub-region.

    Regarding the normalized parameters FGE and CC, both dust emission and transport sub-regions present values with significant stability throughout the year. This actually reveals a small degree of dependence between MONARCH performance and coarse DOD seasonality. In particular, the lowest FGE seasonality can be found in regions with strong seasonal changes in





coarse DOD, namely in NorAfr ($0.45 \leq$ FGE $\leq 0.56$) and MidEas ($0.43 \leq$ FGE $\leq 0.51$). Similarly, seasonal CC values exhibit intra-annual stability in most of the source and transport regions, such as NorAfr ($0.79 \leq$ CC $\leq 0.84$), MidEas ($0.72 \leq$ CC $\leq 0.80$), TroAtl ($0.74 \leq$ CC $\leq 0.82$), AraSea ($0.71 \leq$ CC $\leq 0.76$) and MedSea ($0.72 \leq$ CC $\leq 0.74$). On the other hand, in northern regions where the number of MIDAS observations is lower compared to southern regions, and AEROIASI's contribution to the combined product increases, the seasonal CC values decrease and significant seasonality is noted. This should not be attributed

to any MONARCH uncertainties but to the low quality of AEROIASI retrievals in northern latitudes, as it was concluded in Sect. 3 (Fig. 3), which can bias the regional results of the comparison.

Overall, the seasonal scores of MONARCH total and coarse DOD derived from the comparison with MIDAS+MISR (Fig. S4) and MIDAS+IASI (Fig. S8) respectively, exhibit a similar degree of seasonality by region and by statistical parameter. In regions most affected by dust the results are comparable, with best agreements on CC seasonal scores, implying that the

performance of the reanalysis for both total and coarse DOD is consistent.

## 5   Conclusions

MONARCH dust reanalysis is an advanced dust decadal (2007–2016) regional reanalysis, based on the weather–aerosol–chemistry MONARCH model, providing a continuous 3D representation of atmospheric desert dust over the NAMEE region, with high spatial and temporal resolution. Providing thorough information on dust variations and trends, this product can

be exploited for the development of climate services tailored to key socio–economic sectors, focusing on those that can be significantly affected by atmospheric dust, such as health, transportation and solar energy industry. Therefore, the assessment of the reanalysis performance is of great importance in order to identify its strengths and potential weaknesses, and to be considered in future applications.

Here, we seek to assess the performance of the reanalysis in reproducing the total and coarse DOD using dust products

derived from MODIS, MISR and IASI space-borne instruments along with ground-based remote-sensing measurements from AERONET. Instead of using a single dataset as reference, in the present analysis different observational-based products were combined, which together provide better coverage of the model's spatiotemporal domain. However, each satellite sensor has its own strengths, limitations, and uncertainties. The total and coarse DOD products of the reference datasets (i.e., MIDAS, MISR, and AEROIASI) were obtained following different retrieval techniques and assumptions; limitations on each dust

characterization technique introduce uncertainties into the DOD products. Therefore, an additional advantage of using different observational reference datasets is the ability to perform cross-validation of the model's performance, based on the results obtained from each dataset. By collating the comparison results obtained from the different datasets we can identify biases caused by retrieval uncertainties and assess their contribution to the evaluation results.

Moreover, prior to the reanalysis assessment, we checked the quality of the satellite-based dust products by applying col-

located inter-comparison among the different datasets, using AERONET observations as the reference dataset. Significant discrepancies between satellite and ground-based products over certain regions should be considered as a potential source of skewness for the subsequent model assessment. More specifically, MIDAS and MISR tend to underestimate total DOD in areas





close to the dust sources and slightly overestimate it in remote regions, whereas MIDAS underestimates coarse DOD everywhere; however, both MIDAS and MISR exhibit high correlation (CC > 0.8) and low relative bias (FGE < 0.4) at most sites

where high dust concentrations are recorded (latitudes < 40° N). Lastly, AEROIASI shows moderate correlations (0.4 < CC < 0.8) for stations south of 40° N, but overestimates coarse DOD (MB > 0.07) in remote regions (latitudes > 40° N), showing a weakness in capturing the temporal variations of coarse DOD in these areas at annual or seasonal scales, as indicated by the low correlation with AERONET.

Taking into account these outcomes, the MONARCH reanalysis assessment was based on the comparison against AERONET

and the satellite products, highlighting the similarities among the obtained results for drawing safer conclusions. According to our findings, the MONARCH reanalysis reproduces very well the spatial distribution of atmospheric dust across the NAMEE region, identifying the major dust emission hotspots located in the Sahara Desert and the Middle East, and the main dust transport pathways toward the adjacent maritime regions of the Atlantic Ocean, the Arabian Sea and the Mediterranean Sea. Moreover, the reanalysis is able to reproduce very well the total and coarse DOD seasonal variability with good accuracy, es-

pecially over the aforementioned areas (annual and seasonal CC values consistently greater than 0.6 and up to 0.87), indicating that the reanalysis captures quite well the annual dust cycle both over the sources and the nearby outflow regions.

Quantitatively, according to the evaluation scores, the MONARCH reanalysis seems to simulate more emitted and less transported dust particles. The comparison with the satellite multi-sensor products at regional scale shows that on average, the reanalysis produces slightly higher DOD values over Africa and lower DOD over the Atlantic Ocean and the Arabian Sea.

More specifically, the maximum annual overestimation in total DOD was found in NorAfr (MB = 0.04), the sub-region which contains the Sahara Desert, which is rather insignificant compared to the mean DOD value obtained there from the satellite sensors (MIDAS+MISR DOD = 0.25). Similarly, the MONARCH reanalysis simulates higher coarse DOD over the Sahara Desert compared to the multi-sensor product (MIDAS+IASI); however, the accuracy of this outcome remains questionable due to MIDAS systematic underestimations of the coarse DOD product. On the other hand, the minimum negative MB was

recorded over the maritime regions of TroAtl and AraSea, both for total DOD (MB = −0.04) and coarse DOD (MB = −0.02), indicating a slight underestimation in simulating the exact transported dust quantity in the main downstream directions during the study period. Finally, as an exception to that general conclusion, a near-zero DOD MB was recorded over the MedSea and a near-zero coarse DOD MB over WesAsi, which are considered as dust transport and dust source region, respectively.

The calculation of the FGE, which corresponds to the absolute relative bias of the model, and the CC, which represents

the spatiotemporal correlation between the simulations and the multi-sensor products, shows that the reanalysis performs better (low FGE and high CC) over dust sources and over areas frequently affected by dust transport, whereas the reanalysis scores diminish towards the remote regions located in the northern parts of the study region, where very low annual DOD values are recorded. Both statistical parameters are normalized, allowing comparisons between the total DOD and coarse DOD assessment results. In fact, the annual FGE calculated for each sub-region presents small differences between total and coarse

DOD, whereas the CC is almost identical, especially in regions of high dust concentrations, indicating that the regional scores depend more on the region and its dust levels than on the evaluated variable. According to our regional results, the sub-regions of NorAfr, MidEas, AraSea and TroAtl provide FGE lower than 1 and CC higher than 0.78 for both variables. These good





results are corroborated by the high number of available observations used for the reanalysis assessment in these sub-regions. On the other hand, the remote sub-regions of NorEur and Russia are characterized by very large errors (FGE > 1.93) and low
correlation (CC < 0.6). In this case, the very low availability of MIDAS observations and the significant overestimations of AEROIASI products over these regions prevents us from drawing strong conclusions. To sum up, the MONARCH reanalysis is very reliable over all the regions of frequent dust activity and high dust concentrations where the best normalized statistics (low FGE and high CC) are presented and coincide with large N values, indicating the consistency of the reanalysis when compared against a large number of observations and consequently its very good performance.

The present work shows that using data from different sensors increases the observational coverage, allowing one to assess a larger sample of model data and get better representativeness. More importantly, through the synergy of satellite sensors that perform differently depending on weather conditions, surface type and atmospheric dust concentration, it is possible to better assess the performance of modeling products in conditions where the sensitivity of one sensor to dust particles is higher than another. In this direction, satellite missions like NASA's EMIT (Earth Surface Mineral Dust Source Investiga-
tion; https://earth.jpl.nasa.gov/emit, last access: 14 September 2022) instrument or ESA's EarthCARE (Earth Cloud, Aerosol and Radiation Explorer; https://earth.esa.int/eogateway/missions/earthcare, last access: 14 September 2022) satellite with active sensors on board, in conjunction with improving observational capabilities from the ground through regional research infrastructures, e.g., ACTRIS (www.actris.eu, last access: 14 September 2022), and international initiatives such as GALION (WMO, 2007), could contribute to overcoming current limitations.

*Data availability.* The MONARCH reanalysis dataset (Di Tomaso et al., 2021) is available at http://hdl.handle.net/21.12146/c6d4a608-5de3-47f6-a004-67cb1d498d98 (last access: 14 September 2022). The MIDAS dataset (Gkikas et al., 2020) is available at https://doi.org/10.5281/zenodo.424410 AERONET Version 3 data are available from the AERONET web site (https://aeronet.gsfc.nasa.gov, last access: 14 September 2022). The MISR standard data products can be found at the NASA Langley Atmospheric Data Center (ASDC) Distributed Archive Center (DAAC) (https://asdc.larc.nasa.gov, last access: 14 September 2022). AEROIASI data can be provided upon request to the principal investigator of
the satellite data: Juan Cuesta, LISA/UPEC, cuesta@lisa.ipsl.fr.

**Appendix A: Evaluation metrics**

The evaluation metrics that were used to quantify the performance of the reanalysis products ($M_i$) versus the observational-based retrievals ($O_i$) are presented in Table A1.

where:

$$\overline{M}_d = \frac{1}{N_d} \cdot \sum_{i=1}^{N} M_i \qquad (A1)$$



**Table A1.** Summary of the statistical metrics that were used in the model evaluation.

| Statistic parameter | Equation | Range | Perfect score |
|---|---|---|---|
| Mean Bias | $\text{MB}_d = \overline{M}_d - \overline{O}_d$ | $-\infty$ to $+\infty$ | 0 |
| Root Mean Square Error | $\text{RMSE}_d = \sqrt{\frac{1}{N_d} \cdot \sum_{i=1}^{N}(M_i - O_i)^2}$ | 0 to $+\infty$ | 0 |
| Fractional Gross Error | $\text{FGE}_d = \frac{2}{N_d} \cdot \sum_{i=1}^{N}\left|\frac{M_i - O_i}{M_i + O_i}\right|$ | 0 to 2 | 0 |
| Correlation Coefficient | $\text{CC}_d = \frac{\sum_{i=1}^{N}(M_i - \overline{M}_d) \cdot (O_i - \overline{O}_d)}{\sqrt{\sum_{i=1}^{N}(M_i - \overline{M}_d)^2} \cdot \sqrt{\sum_{i=1}^{N}(O_i - \overline{O}_d)^2}}$ | $-1$ to 1 | 1 |

$$\sigma M_d = \sqrt{\frac{1}{N_d} \cdot \sum_{i=1}^{N}(M_i - \overline{M}_d)^2} \qquad (A2)$$

$$\overline{O}_d = \frac{1}{N_d} \cdot \sum_{i=1}^{N} O_i \qquad (A3)$$

$$\sigma O_d = \sqrt{\frac{1}{N_d} \cdot \sum_{i=1}^{N}(O_i - \overline{O}_d)^2} \qquad (A4)$$

are the mean and the standard deviation of the reanalysis (Eq. A1–A2) and the observed DOD (Eq. A3–A4); N indicates

the total number of collocated and concurrent $M_i$-$O_i$ pairs. The subscript d denotes the observational dataset used in the calculations.

MB captures the average deviation between the two datasets. Negative/positive MB indicates underestimation/overestimation of the reanalysis with respect to the observations. It theoretically ranges from $-\infty$ to $+\infty$ and its perfect score is 0.

RMSE represents the root mean square difference between the reanalysis and observations. It is a measure of how spread out

these differences are. RMSE is strongly dominated by the largest differences due to the squaring operation. It ranges between 0 and $+\infty$ and its perfect score is 0.

FGE is a measure of the mean absolute relative bias where the difference between reanalysis and observation is normalized by their mean value. It is a positively defined indicator that behaves symmetrically with respect to under- and overestimation, without over emphasizing outliers. FGE ranges from 0 to 2 (i.e., from 0 to 200 %), where 0 indicates a perfect agreement and

values close to 1 or greater indicate very poor agreement.

CC indicates the extent to which spatial and temporal patterns in the reanalysis match those in the observations, quantifying their correlation and dependence. It ranges between $-1$ and 1, where $-1$ means perfect anti-correlation, 0 means no correlation, and 1 indicates perfect correlation.

The statistical results obtained by the comparison between the reanalysis and each reference satellite dataset can be aggre-

gated in order to get total average scores by weighting the metrics of Table A1 by the number of observations $N_d$ provided by each dataset, using the equations of Table A2.





**Table A2.** Weighted mean of the evaluation metrics obtained by different reference satellite datasets.

| Statistic parameter | Equation |
|---|---|
| Mean Bias | $\mathrm{MB} = \frac{\sum_{d=1}^{2} N_d \cdot \mathrm{MB_d}}{\sum_{d=1}^{2} N_d}$ |
| Root Mean Square Error | $\mathrm{RMSE} = \sqrt{\frac{\sum_{d=1}^{2} N_d \cdot \mathrm{RMSE_d^2}}{\sum_{d=1}^{2} N_d}}$ |
| Fractional Gross Error | $\mathrm{FGE} = \frac{\sum_{d=1}^{2} N_d \cdot \mathrm{FGE_d}}{\sum_{d=1}^{2} N_d}$ |
| Correlation Coefficient | $\mathrm{CC} = \frac{\sum_{d=1}^{2} N_d \cdot [\mathrm{CC_d} \cdot \sigma M_d \cdot \sigma O_d + (\overline{M}_d - \overline{M}) \cdot (\overline{O}_d - \overline{O})]}{\sqrt{\sum_{d=1}^{2} N_d \cdot [\sigma M_d^2 + (\overline{M}_d - \overline{M})^2]} \cdot \sqrt{\sum_{d=1}^{2} N_d \cdot [\sigma O_d^2 + (\overline{O}_d - \overline{O})^2]}}$ |

where:

$$\overline{M} = \frac{\sum_{d=1}^{2} N_d \cdot \overline{M}_d}{\sum_{d=1}^{2} N_d} \tag{A5}$$

$$\sigma M = \sqrt{\frac{\sum_{d=1}^{2} N_d \cdot [\sigma M_d^2 + (\overline{M}_d - \overline{M})^2]}{\sum_{d=1}^{2} N_d}} \tag{A6}$$

$$\overline{O} = \frac{\sum_{d=1}^{2} N_d \cdot \overline{O}_d}{\sum_{d=1}^{2} N_d} \tag{A7}$$

$$\sigma O = \sqrt{\frac{\sum_{d=1}^{2} N_d \cdot [\sigma O_d^2 + (\overline{O}_d - \overline{O})^2]}{\sum_{d=1}^{2} N_d}} \tag{A8}$$

are the weighted mean and the combined standard deviation of the reanalysis (Eq. A5–A6) and the aggregated satellite-based
dust products (i.e., MIDAS+MISR for DOD and MIDAS+AEROIASI for coarse DOD) (Eq. A7–A8). The subscript d denotes
the satellite dataset used in the calculations (d = 1 MIDAS and d = 2 MISR for DOD; as well as d = 1 MIDAS and d = 2
AEROIASI for coarse DOD).

*Author contributions.* MM, LM, SB and EDT designed the study and the whole analysis. SB collected and prepared the datasets. MM
processed all the datasets and performed the data analysis. SC contributed to the statistical analysis of the data. MM wrote the initial
manuscript with contributions from LM and ST. MM, LM, SB, EDT and CPGP discussed the main results. RK, SB, EDT, CPGP, AG, JC,
ST, CD and OJ reviewed and edited the manuscript. RK contributed also to the part of the paper relating to MISR data. EDT contributed also
to the part relating to the MONARCH reanalysis data. AG contributed also to the part relating to MIDAS data. JC and PF contributed also to
the part relating to AEROIASI data. All co-authors contributed to the final editing of the manuscript. LM supervised the entire work.





*Competing interests.* The authors declare that they have no conflict of interest.

*Acknowledgements.* The authors thank all the Principal Investigators and their staff for establishing and maintaining the NASA and PHO-
TONS AERONET sites, and the MODIS and MISR mission scientists and associated NASA personnel for the production of the data used
in this study. BSC co-authors acknowledge PRACE (eDUST, eFRAGMENT1 and eFRAGMENT2) and RES (AECT-2019-3-0001, AECT-
2020-1-0007, AECT-2020-3-0013) for awarding access to MareNostrum at the BSC and for providing technical support.

*Financial support.*

This research has been supported by the DustClim project which is part of ERA4CS, an ERA-NET programme initi-
ated by JPI Climate with co-funding by the European Union's Horizon 2020 research and innovation programme (Grant
no 690462). The authors acknowledge the ACTRIS-IMP (Implementation project), funded by the European Union's Hori-
zon 2020 research and innovation programme (Grant no 871115). Michail Mytilinaios, Sergio Ciamprone and Claudio Dema
have received funding from CIR01_00015 - PER-ACTRIS-IT "Potenziamento della componente italiana della Infrastruttura
di Ricerca Aerosol, Clouds and Trace Gases Research Infrastructure-Rafforzamento del capitale umano" - Avviso MUR D.D.
n. 2595 del 24.12.2019 Piano Stralcio "Ricerca e Innovazione 2015–2017". BSC co-authors acknowledge support from the
European Research Council under the European Union's Horizon 2020 research and innovation programme (grant n. 773051;
FRAGMENT), the AXA Research Fund (AXA Chair on Sand and Dust Storms), and the contribution agreement between
AEMET and BSC to carry out development and improvement activities of the products and services supplied by the World
Meteorological Organization (WMO) Barcelona Dust Regional Center (i.e., the WMO Sand and Dust Storm Warning Advisory
and Assessment System (SDS-WAS) Regional Center for Northern Africa, the Middle East and Europe). The work of Ralph
Kahn is support in part by the NASA Aerosol-Cloud Modeling and Analysis Program under Richard Eckman, and the NASA
Earth Observing System Terra and MISR projects. Antonis Gkikas acknowledges support by the Hellenic Foundation for Re-
search and Innovation (H.F.R.I.) under the "2nd Call for H.F.R.I. Research Projects to support Post-Doctoral Researchers"
(Project Acronym: ATLANTAS, Project Number: 544). The MIDAS dataset has been developed in the framework of the
DUST-GLASS project (grant no. 749461; European Union's Horizon 2020 Research and Innovation programme under the
Marie Skłodowska-Curie Actions). The AEROIASI product is developed at the LISA laboratory with the financial support of
the IASI-TOSCA (Terre, Océan, Surface Continentale et Atmosphère) project from the Centre National des Etudes Spatiales
(CNES) and technical assistance of the AERIS French national data centre. IASI is a joint mission of EUMETSAT and CNES.





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
