# Peer review of "Comparison of dust optical depth from multi-sensor products and the MONARCH dust reanalysis over Northern Africa, the Middle East and Europe"

_Atmospheric Chemistry and Physics, 2022_

## Referee Comment (RC2)

**Review of ACP manuscript acp-2022-655 "Comparison of dust optical depth from multi-sensor products and the MONARCH dust reanalysis over Northern Africa, the Middle East and Europe" by Michail Mytilinaios, Sara Basart, Sergio Ciamprone, Juan Cuesta, Claudio Dema, Enza Di Tomaso, Paola Formenti, Antonis Gkikas, Oriol Jorba, Ralph Kahn, Carlos Pérez García-Pando, Serena Trippetta, and Lucia Mona**

General comments

This paper presents a thorough assessment of a novel atmospheric dust reanalysis dataset with unique characteristics (high resolution, dust properties). The assessment uses a combination of different satellite datasets (with different strengths and weaknesses, and with different information content) and ground-based observations (with limited spatial sampling). The combination of different reference datasets allows a meaningful assessment despite of the given limitations of each dataset. The analysis and the conclusions are thoroughly made and presented and altogether they support the high quality of the evaluated reanalysis dataset.

The paper discusses a relevant topic (assessing the quality of a novel atmospheric dust reanalysis), since atmospheric dust has several strong impacts on the Earth system, human health and economic sectors. The comparison methodology applied is clearly outlined and follows the state of the art. Related work of the authors and the community are properly credited (while I note that the list of references is quite long). Title and abstract as well as the language (as far as I can judge as non-native English speaker) are well suited. The quantities used to assess the dataset quality are clearly defined in the Appendix. The authors have made a meaningful selection of the multiple elements in the study for the main paper and added the additional material in a supplement (the parts of which are also referenced in the main paper).

The only significant improvement which I suggest, is an enhancement of guidance for a reader through the vast collection of evaluation steps done (see my first specific comment). This means a major revision (but likely not too high efforts) in terms of the paper presentation / structure, but only minor revisions in terms of its content.

Specific comments

My main point is on the presentation / structure of the paper: The paper presents a rich set of analysis to assess the quality of the MONARCH reanalysis dataset. The authors have already made an effort to avoid overwhelming the reader by splitting of parts of the material into a supplement. However, even in the main paper as it is now a reader may easily loose the overview with so many different aspects. I therefore recommend for better guidance of the reader:

- to provide an overview table in the beginning of section 2 or in section 2.6 which lists the different analysis made (e.g. satellite validation, reanalysis assessment of DOD, coarse DOD, annual, seasonal, …) versus the various reference datasets used; this should include the analysis shown in the supplement and at the same time identify the placing in the paper (main paper or supplement) for each analysis element
- to add sub or sub-sub headings (e.g. for the section on DOD and coarse DOD) in the analysis and discussion sections
- to add short titles to the figures to help a quick reader

Further specific comments:

Some paragraphs do not seem to fit in the section they are presented in – they should be moved:

- lines 274 – 287: should better become an overview at the start of section 2 or in section 2.6 (and there the overview table which I suggested could be added, too)
- lines 347 – 357: should better be moved to the discussion section

In the abstract the tense is mixed up (mostly present, but at some places simple past is used, e.g. line 11) – this should be harmonized.

In the abstract (line 24/25) relative bias are provided as fractions of 1, which can easily be mixed up with absolute AOD values. I therefore recommend to use %-values.

In the introduction impact of atmospheric dust is summarized; I miss a statement on fertilizing the Amazonas here.

In summarizing the MONARCH reanalysis in the introduction, data assimilation of specific satellite datasets is mentioned. Can you add here which datasets are used, so that a reader can understand immediately in how far these are different from the datasets which are used for the evaluation in this paper.

Section 2.4: can you add a statement, why you chose this particular IASI dataset and what unique characteristics it has as compared to the IASI datasets provided by the operational Copernicus Climate Change Service / Climate Data Store (https://cds.climate.copernicus.eu/cdsapp#!/dataset/satellite-aerosol-properties?tab=overview).

Figures 2 – 5: The top row images (2 per column) are too small to be able to see much in them (on paper) – please enlarge them both to the size of the other maps and place them one below the other on top of each column (this should still fit on one page)

In many places you alternate "MONARCH" with "reanalysis" – for reader guidance I would find it easier to follow, if you keep "MONARCH" in all places

Fig. 4 and 5: There is a strong land-sea contrast I the MB map for MIDAS, which I missed in the text – can you please add this

Technical corrections
Line 52: delete "the" before "airborne aerosols"
Line 76: better replace "data availability" by "measurement possibility"
Line 79: better replace "e.g.," by "and" (as there are two separate aspects in this sentence)
Line 19/20: I do not understand the statement iii) – can you please extend or reword to explain what this means?
Line 131: better replace "-30° E" by "30° W"
Line 151: you could add "profile" before "variables"
Fig. 1: should the area "Northern Europe" not better be renamed to "Northern and Central Europe"?
Line 175: can you add some quantitative information here (an average 5 value for example)
Lines 178 – 187: I find this discussion of technical matching in UTC confusing, as the difference in local time is smaller – maybe you should point this out in addition
Lines 244 and 260: the radius limits of 2 μm and 0.6 μm differ – can you please explain?
Line 282: nighttime measurements: are they used at all in this paper? Please state explicitly.
Lines 284/285: "usually" and "in most cases" duplicate their meaning

Lines 300/301: this means that these cases are not used at all, correct? Please state this explicitly

Line 303: fine mode radius of 0.6 μm – you should mention / discuss the impact of the different definition elsewhere (0.5 μm; e.g., line 57)

Line 304: can you mention that wildfires may lead to significantly high AOD values (versus sea salt with only low values)?

Lines 317/318: I do not understand this sentence, can you please reword?

Line 320: can you please add the number of stations which meet this criterion?

Lines 505 – 507: can you add a reference for this statement?

Line 709: please add "total column"

---

## Author Comment (AC1)

Dear Editor, dear Referees and Reviewers,

first of all, we would like to thank you for accepting our manuscript for public peer review in the ACPD forum and for considering it for publication in ACP. We are especially grateful to the anonymous reviewers for the detailed comments and suggestions during the acceptance and the discussion phase which greatly improved the quality of our manuscript; we sincerely appreciate their contribution a lot.

Here we present our responses to all the comments posted by the Anonymous Referee #2 (acceptance phase) and by the two Anonymous Reviewers (#2 and #3; discussion phase). All the comments are repeated here in *italic*, and where is needed we report the modified text as "`Revised text`", where the text added in the original submitted manuscript is underlined and the text deleted is .

Lastly, in the revised manuscript we have also updated the affiliations and the date of last access when citing websites, as well as the "Author contributions" and the "Acknowledgements" sections.

**Authors' response to Anonymous Referee #2**

*Why are the authors dismissive of the MISR over-land products? See, e.g., Kahn et al., dot: 10.1002/2015JD023322 (2015) for MISR aerosol type classification which shows retrievals over the land are successfully captured. I would think MISR's discrimination of dust versus non-dust is at least as good as MODIS.*
*Otherwise the paper seems suitable for pre-access.*

We would like to thank Referee #2 for their reasonable question, which gives us the opportunity to explain our choice and improve the corresponding passage in our manuscript.

First of all, we know from the literature that MISR Dark-Water (DW) algorithm is very reliable in distinguishing non-spherical particles (Kalashnikova and Kahn, 2006), whereas the Heterogeneous-Land (HL) algorithm has greater uncertainty especially over bright surfaces. In addition, we've noticed that over the outflow regions the MISR DW DOD values are higher than the HL DOD (Fig. A). For example, higher DOD values are observed in the Atlantic Ocean, the Gulf of Guinea, the Persian Gulf, the Red and the Mediterranean Sea than over the adjacent coastal areas where the dust plume is actually coming from. This behavior is less expected, considering that a 10-year average DOD should decrease with distance from the sources, due to deposition mechanisms. This inconsistency appears to be due to an underestimation of the DOD values retrieved by the HL algorithm. Moreover, the great uncertainty of MISR over-land retrieval also results from the Coefficient of Variation (CV) map, which is defined as the ratio of the standard deviation to the mean DOD (Fig. B). Consequently, to avoid misleading results in the model assessment, we decided to use only the DW retrievals.

This in no way implies that the MISR dust product is of lower reliability than the other two satellite datasets. In fact, not only does the AOD retrieval sensitivity decrease with the brightness for almost all satellite sensors – something already mentioned in the submitted manuscript –, but also MISR's ability (as a multi-angle instrument) to discriminate dust from other aerosol types is better than single-view instruments like MODIS.

However, in this study, except for MISR, we are not using products obtained directly from MODIS or IASI measurements, but datasets that were based on those products, i.e., MIDAS and AEROIASI respectively. In the submitted manuscript we present the uncertainties of both datasets and even perform our own comparison with AERONET, specifically for the temporal and spatial domain of interest. However, the distinction between areas of low and high uncertainty was not a matter of a single factor, as it was for MISR (i.e., surface type), in fact, the MIDAS land-water uncertainty difference is insignificant. Therefore, it was decided to include the entire datasets in the model assessment and to specify in the manuscript the areas where uncertainties may affect the comparison results.

[Figure]

**Figure A**: Mean MISR DOD for the period 2007–2016

[Figure]

**Figure B**: Mean MISR coefficient of variation for the period 2007–2016

Revised text (lines: 202–205, Section 2.3):

Thus, the AOD fraction of the non-spherical particles, consisting of randomly oriented non-spherical grains or ellipsoids, can be considered equivalent to the DOD with relative certainty, especially over dark-water surfaces (Kalashnikova and Kahn, 2006). Many studies show that MISR's sensitivity to DOD depends on the surface type and  like nearly all passive satellite aerosol remote-sensing, MISR  retrievals are less reliable over bright surfaces (Kahn et al., 2010). Specifically, MISR over-land retrievals tend to underestimate DOD in dust-rich areas and have greater uncertainties compared to MISR over-water DOD; therefore, for the MONARCH DOD comparison we used exclusively dark-water retrievals, which are exceedingly sensitive to aerosol non-sphericity (Guo et al., 2013; Kalashnikova et al., 2013).

Authors' response to Anonymous Reviewer #2
https://doi.org/10.5194/acp-2022-655-RC1

*The paper presents a thorough evaluation of the MONARCH regional dust reanalysis over North Africa, Europe, and Middle East in terms of comparisons to satellite and ground-based dust aerosol optical depth products (DOD). The presentation is well written and the figures—although complex—are relatively easy to read. I have only a few minor clarifications and two major points, though I don't think they require extensive revision.*

We would like to thank Reviewer #2 for their positive feedback on our manuscript. Moreover, their comments not only did allow us to clarify some important findings, but also pointed out some additional ones.

*1) Line 153 - I am confused by the use of the word "ensemble" here and in the remainder of this paragraph. What ensemble is being referred to? The statistics presented later are data sets versus one another of the reanalysis versus the data sets.*

The word "ensemble" here refers to the output (analysis ensemble) of the local ensemble transform Kalman filter data assimilation scheme used for the production of the reanalysis. More specifically, the MONARCH reanalysis has been produced by estimating model uncertainty from the realizations of the dust fields in an ensemble of MONARCH model calculations. Each member was generated using different meteorological initial and boundary conditions, and dust emission schemes, along with additional perturbations in the model emission parameters. A set of ensemble statistics has been calculated and archived for each reanalysis data set output variable, namely the analysis ensemble mean, standard deviation, maximum and median. More details about the whole procedure can be found in Di Tomaso et al. (2022). In our paper we are assessing the MONARCH reanalysis using only the DOD and coarse DOD ensemble mean. We revised the manuscript adding a few words about the ensemble.

Revised text (lines: 156–162, Section 2.1):
The MONARCH reanalysis dataset consists of upper-air profile variables such as dust mass concentration and extinction coefficient at 550 nm, surface fields such as accumulated dust dry and wet deposition and mass surface concentration, and total column fields like instantaneous total column dust load, DOD and coarse DOD at 550 nm. The reanalysis has been produced by estimating model uncertainty from the realizations of the dust fields in a 12-member ensemble, where each ensemble member was generated using different meteorological initial and boundary conditions and dust emission schemes, along with additional perturbations in the model emission parameters. For each variable of the reanalysis a number of ensemble statistics is available Calculation of basic ensemble statistics was performed for each reanalysis variable, namely the ensemble arithmetic mean, standard deviation, median and maximum of the ensemble members.; however, iIn this the present paper we assess exclusively the reanalysis ensemble mean, as it is a more representative value than the median for describing the ensemble, as it considers all the members of the ensemble without excluding the outliers.

*2) Line 402 - I don't agree "low RMSE" in the region being described. I see high RMSE in bottom left quadrant of, e.g., Figure 2j.*

The reviewer is right. Thank you for this remark! We actually did not refer in the manuscript to the stations located near the Gulf of Guinea where indeed MIDAS RMSE presents its maximum values. We modified the text by adding more information regarding the RMSE distribution near to the dust sources.

Revised text (lines: 404–407, Section 3.1):
As in the case of MB, MIDAS presents maximum RMSE near to the coasts of the Gulf of Guinea (> 0.32; Fig. 2j); however, RMSE is relatively low (< 0.24) along the Sahelian Belt and across the Arabian Peninsula, considering the high DOD values (> 0.32) observed there by both AERONET and MIDAS. This also applies to MIDAS FGE that remains low (< 0.7) in regions of high dust activity (Fig. 2m). At stations located in the western Sahara Desert and the western part of the Sahelian Belt MIDAS shows low RMSE (up to 0.12; Fig. 2j) and FGE (up to 0.7; Fig. 2m) considering the high DOD values (annual DOD mean above 0.36 for both MIDAS and AERONET).

*3) Line 406 - I don't understand MISR shows smaller differences against AERONET in the regions indicated. It looks worse than MIDAS in terms of MB on the North African coast (comparing Figure 2g and 2h).*

Exactly! MISR shows larger differences against AERONET than MIDAS in the regions indicated. We actually wrote MB < -0.1 that in terms of absolute value means |MB| > 0.1, which is larger than MIDAS |MB|. We rephrased the sentence to clarify what we wanted to say there.

```
Revised text (lines: 408-410, Section 3.1):
```
MISR MB shows an overall underestimation at the majority of the AERONET sites, which exceeds -0.07 in the surroundings of the dust sources. The largest underestimations can be found along the Northern Africa coastline and in the Red Sea, where MB < -0.1 (Fig. 2h).

*Two major comments:*
*(I think the authors should address these two major points in their conclusions at least with a couple of paragraphs that expand on these ideas as the pertain to their study.)*

*4a) I question the conclusion stated on line 737. The "gold standard" here is AERONET, and the comparisons in Section 3 make plain that MIDAS and AEROIASI underestimate AERONET AOD over source regions in both total and coarse DOD (MIDAS) and coarse DOD (AEROIASI). MONARCH is high AOD over those regions relative to MIDAS and AEROIASI, but still low compared to AERONET in both total and coarse DOD (Figure S3 and S7). Admittedly you don't have the regional coverage from AERONET that the satellites provide, but still I think my assessment there is fair.*

The conclusion we stated (line 778 of the revised manuscript: "Quantitatively, according to the evaluation scores, the MONARCH reanalysis seems to simulate more emitted and less transported dust particles.") is based on detailed analysis presented in the paper. We try in the following to make it clear addressing the different related points raised by the reviewer:

I) MIDAS and AERONET total DOD are actually in a very good agreement over the dust source regions.

Considering the comparison made between MIDAS and AERONET total DOD, MIDAS shows very good scores at stations close to the dust sources (Section 3.1 of the revised manuscript). In Fig. C we reproduce the 1st column of Fig. 2 of the manuscript only for stations where the AERONET DOD > 0.1 (i.e., sites of frequent dust events). In fact, MIDAS DOD shows very low biases and errors and high correlation in almost all the stations except for those three stations near to the coast of the Gulf of Guinea, something that we have already mentioned in the submitted manuscript (lines: 401–402 of the revised manuscript, Section 3.1).

Moreover, if we take a closer look at the stations that reside within the sub-region of Northern Africa (as defined in Fig. 1 of the manuscript), the results are even better. In fact, the average scores of 13 stations listed in Table A show that MB is insignificant (-0.03) and RMSE is very low (0.14) compared to the DOD observed by MIDAS (0.28) and AERONET (0.31), FGE is very low considering that its scale goes from 0 to 2, and lastly CC is remarkably high (0.90). Consequently, we can safely assess that MIDAS total DOD is highly trustworthy near the dust sources and especially in this sub-region.

[Figure]

**Figure C**: MIDAS vs AERONET DOD only at sites that are affected frequently by dust events (mean AERONET DOD > 0.1)

| Northern Africa (2007-2016; 13 sites) | MIDAS | AERONET |
|:---:|:---:|:---:|
| **DOD** | 0.28 ± 0.14 | 0.31 ± 0.14 |
| **MB** | -0.03 ± 0.04 ||
| **RMSE** | 0.14 ± 0.04 ||
| **FGE** | 0.53 ± 0.25 ||
| **CC** | 0.90 ± 0.07 ||

**Table A**: Mean regional scores (± standard deviation) of MIDAS DOD compared to AERONET, computed from the validation results at 13 AERONET stations located in the sub-region of Northern Africa.

II) MONARCH DOD is usually lower than AERONET DOD near the dust sources, but only over the AERONET stations, which are not representative of the whole region. The visualizations and the mean regional values of the MONARCH DOD scores can mislead.

The reviewer questions the conclusion: "Quantitatively, according to the evaluation scores, the MONARCH reanalysis seems to simulate more emitted and less transported dust particles.", noting that "*MONARCH is high AOD over those regions relative to MIDAS and AEROIASI, but still low compared to AERONET in both total and coarse DOD (Figure S3 and S7).*". Indeed, Fig. 4g of the manuscript shows that in most parts of Northern Africa MONARCH DOD > MIDAS DOD, whereas in Fig. 4i we see that MONARCH DOD < AERONET DOD, at most stations of that region. The same conclusion can be drawn if we collate the regional values of NorAfr in Fig. S1 of the supplement (annual MB = 0.04) with the values in Fig. S3 (annual MB = -0.06). However, these results are not exactly comparable. MIDAS and AERONET have different spatial and temporal coverage, consequently they evaluate different sub-set of the MONARCH dataset. In fact, the MIDAS regional values (Fig. S1) represent thousands of grid points (pixels), whereas the AERONET regional values are derived from few stations, unevenly distributed across a region, which in turn represent only a tiny spot on the map, rather than tens of kilometers covered by the colored dots that we used on the AERONET maps for display purposes only. In the submitted manuscript (lines: 520–530 of the revised manuscript, Section 4.1.1) we provided more arguments trying to explain the discrepancies between the MONARCH regional scores derived from the comparison with AERONET and the satellites.

III) The MONARCH evaluation scores over AERONET stations derived from the comparison with MIDAS and AERONET are actually quite similar.

Here we make a straight comparison between MONARCH vs MIDAS MB and MONARCH vs AERONET MB at station level. In Fig. D we are using the results of Fig. 4g of the manuscript and we display only the pixels (pixel area ~ 10 km x 10 km) that encloses the AERONET stations (where the AERONET DOD > 0.1). Now it is not clear anymore that MONARCH DOD > MIDAS DOD in Northern Africa because at most of the sites MB < 0. Moreover, the two maps of Fig. D are in a very good agreement. Particularly, all of the red dots on the MONARCH vs AERONET map (Fig. D bottom map) can also be found on the other map, and any quantitative differences that arise in the remaining Northern Africa sites are statistically insignificant. In fact, the average MONARCH MB of 14 sites located within Northern Africa equals to 0.01 ± 0.05 when compared to MIDAS, and it is -0.06 ± 0.07 when compared to AERONET (see plot on the right in Fig. D).

This is a confirmation that the results obtained only at AERONET sites over source regions (slight underestimation of MONARCH) is not geographically representative of the whole source region and shows how much is important to extend the model reanalysis evaluation over a dataset more extended geographically respect to the AERONET golden standard. In conclusion, we can safely state that the MIDAS over land and the MIDAS+MISR over water total DOD datasets are reliable and that the reanalysis evaluation results from extensive areas, where no AERONET observations are available, are trustable.

IV) The MONARCH MB results are questionable for coarse particles.

The reviewer's argument about satellites coarse DOD is valid. It is stated in the submitted manuscript several times that MIDAS and AEROIASI underestimate coarse DOD over the source regions (e.g., lines: 712 and 763 of the revised manuscript, Sections 4.2.2 and 5), therefore the MONARCH MB results are questionable for coarse particles (line: 786 of the revised manuscript, Section 5). Indeed, differences in how each coarse DOD dataset is defined and retrieved can introduce large uncertainties in the reanalysis assessment. For example, as noted in the manuscript, MIDAS uses a smaller cut-off radius (0.5 μm) than MONARCH, AERONET and AEROIASI (0.6 μm), additionally AEROIASI is more sensitive to particles with radius > 1 μm, whereas AERONET coarse DOD can be contaminated by sea-salt especially in coastal areas.

[Figure]

**Figure D**: MONARCH vs MIDAS MB (top left) and MONARCH vs AERONET MB (bottom left) only at sites where AERONET stations are located and where AERONET DOD > 0.1. Mean regional MONARCH MB (± standard deviation) (right), computed from the MB results against MIDAS and AERONET at 14 sites located in the sub-region of Northern Africa.

*4b) Comparing the numbers in Figure S3 and S7 for North Africa, there is also some insight into the coarse/total DOD partitioning in the model and data, as well as interesting aspects in seasonal variability (MONARCH goes from a ratio of 0.14:0.4 (35%) to 0.22:0.31 (71%) from MAM to JJA; AERONET goes from 0.19:0.48 (40%) to 0.26:0.39 (67%)). That's pretty good and that seasonal change in the coarse:total DOD ratio seems like something to note. The authors are well aware of the recent body of literature on the "under appreciated" dust coarse mode, and MONARCH seems to be doing something robust there.*

We would like to thank the reviewer for pointing out this aspect. We revised the manuscript accordingly.

```
Revised text (lines: 569-587, Section 4.1.2):
```
Regarding the MONARCH versus AERONET comparison at regional and seasonal scale (Fig. S7), the typical patterns in seasonality are also found here. At AERONET sites located close to the dust sources (NorAfr and MidEas) and in their outflow regions (TroAtl and AraSea), the MONARCH reanalysis correctly provides maximum coarse DOD values in MAM and JJA. Moreover, the MONARCH reanalysis succeeds in identifying the dry season months at the sites south of the Sahel (i.e., DJF and MAM in SubSah). However, MONARCH's annual and seasonal coarse DOD is almost everywhere lower than the values provided by AERONET (MB < 0). This is due to the fact that the AERONET coarse DOD product can be contaminated by other coarse particles as well. The contribution of other aerosols can be insignificant in southern latitudes (< 40° N) where mineral dust is the dominant type, but under low dust conditions their impact increases. In fact, in the remote regions of NorAtl, NorEur and Russia, and in MedSea where sea-salt predominates in coastal stations, AERONET coarse DOD (Fig. S7) results in most seasons greater than AERONET total DOD (Fig. S3), which is impossible. AERONET's coarse DOD overestimates naturally lead to large relative biases compared to MONARCH coarse DOD (FGE > 1.17), and thereby the validity of the MONARCH reanalysis evaluation

results reported in those regions diminishes. On the other hand, in the regions where coarse DOD is higher, the seasonal results of both normalized metrics are very good (FGE < 1 and CC > 0.7) and quite stable. This means that the MONARCH reanalysis reproduces the seasonal variability of coarse DOD very well compared to AERONET, although not in absolute values. In fact, there are many similarities between the seasonal change of MONARCH and AERONET Coarse Mode Fraction (CMF), which is defined as the coarse DOD to total DOD ratio. For example, considering the seasonal DOD and coarse DOD in NorAfr (Figs. S3 and S7) the seasonality of MONARCH CMF (DJF: 26 %; MAM: 35 %; JJA: 70 %; SON: 49 %) is consistent with AERONET's (DJF: 21 %; MAM: 40 %; JJA: 68 %; SON: 44 %). This implies that the MONARCH reanalysis reproduces very efficiently the size distribution of the dust particles at the sites in the vicinity of the Sahara Desert. ~~The seasonal MB and RMSE values over the dust emission and dust transport regions correspond to those obtained by the total DOD comparison. On the other hand, over the remote regions, MB is higher in absolute terms because AERONET tends to overestimate coarse DOD (> 0.03) in northern latitudes (> 40° N), providing greater seasonal and annual values than for total DOD (Fig. S3). Nevertheless, the normalized metrics (i.e., FGE and CC) are overall better for coarse DOD than for total DOD (Fig. S3).~~

*5) On the other hand, the gradient from land to ocean is noted as a discrepancy and there are at least two things there to consider. First, it seems notable that the data assimilation only corrects the model over land using the MODIS Deep Blue products. (An interesting aside: how do the data being assimilated compare to the AERONET data? This seems not to be considered here.) So if the data is correcting forward model biases over land that may be conveyed over ocean. Secondly, to those model biases, that models tend to deplete dust over land range transport too efficiently is I think also well known. Without an evaluation of the dust vertical profile it's hard to tell what's going on here. It's also the case that some models are overly aggressive in loss processes, both dry and wet, and some statement of that to the effect of the budget of MONARCH compared to, say, AEROCOM models would be useful.*

Yes, indeed, only Deep Blue (over-land) MODIS coarse-DOD retrievals were assimilated for the production of the MONARCH reanalysis. The validation of this dataset is out of the scope of our manuscript and comparison between a previous version of this product and AERONET-derived DOD can be found in the cited paper Ginoux et al. (2012b, Section 4.1).

We note that though the satellite DOD retrievals are present only over land, they had some influence on the analysis also over sea (in the proximity of the coast) according to the model background spatial covariance and limited by the observation radius of influence. As the reviewer has pointed out, the forward model biases that are corrected by the assimilation procedure might affect the transport. As shown in Di Tomaso et al. (2022, Figure 6), the largest positive or negative analysis increments obtained during the reanalysis production correspond to areas with the strongest dust load, i.e., to source regions and their vicinity. In particular, the large negative mean analysis increments point to an overestimation of some of the main sources' strength in the forward model (e.g., Bodélé Depression in Chad; in the Saudi Arabia lowlands). This overestimation is corrected by data assimilation by removing mass from the atmosphere, which could affect long-range transport.

Additionally, to having observational constraint essentially only over land, the gradient from land to ocean of the performance of the reanalysis could be attributed also to the forward model depositing dust too efficiently in the transport, as the reviewer correctly says. As shown in Klose et al. (2021, in Table 7) the annual average lifetime of dust in MONARCH is approximately of 3 days (for the three dust emission schemes used in the reanalysis production), which puts MONARCH next to the AEROCOM models with a medium/low life time for dust according to, for example, Huneeus et al. (2011, Table 3). Assimilating observations also over sea would help in this respect, since observations could correct more efficiently possible model deficiencies in the transport and deposition.

A study of the dust vertical structure could indeed help to investigate more in detail the performance of reanalysis and is already under preparation for a companion paper. Part of the co-authors team worked on it and in particular on the evaluation of the dust extinction profiles of the MONARCH reanalysis against ground-based (ACTRIS-EARLINET) and satellite lidar (CALIPSO-LIVAS) profile products. As the reviewer points, model intercomparisons, as the ones considered in AEROCOM, are useful exercises for understanding source of uncertainties and degree of variation between the state-of-the-art atmospheric models. Here it is worthy to mention that AEROCOM considers mostly global models while a regional configuration of the MONARCH model has been used for the production of the reanalysis.

Following the reviewer's comment, we have added the following sentence in Section 5 of the revised manuscript.

```
Revised text (lines: 779-780, Section 5):
```
Quantitatively, according to the evaluation scores, the MONARCH reanalysis seems to simulate more emitted and less transported dust particles. This could be due to several factors such as having observational constraint mostly over land only or potential issues in the dust deposition and transport in the underlying model.

Authors' response to Anonymous Reviewer #3
https://doi.org/10.5194/acp-2022-655-RC2

*General comments*
*This paper presents a thorough assessment of a novel atmospheric dust reanalysis dataset with unique characteristics (high resolution, dust properties). The assessment uses a combination of different satellite datasets (with different strengths and weaknesses, and with different information content) and ground-based observations (with limited spatial sampling). The combination of different reference datasets allows a meaningful assessment despite of the given limitations of each dataset. The analysis and the conclusions are thoroughly made and presented and altogether they support the high quality of the evaluated reanalysis dataset.*

*The paper discusses a relevant topic (assessing the quality of a novel atmospheric dust reanalysis), since atmospheric dust has several strong impacts on the Earth system, human health and economic sectors. The comparison methodology applied is clearly outlined and follows the state of the art. Related work of the authors and the community are properly credited (while I note that the list of references is quite long). Title and abstract as well as the language (as far as I can judge as non-native English speaker) are well suited. The quantities used to assess the dataset quality are clearly defined in the Appendix. The authors have made a meaningful selection of the multiple elements in the study for the main paper and added the additional material in a supplement (the parts of which are also referenced in the main paper).*

*The only significant improvement which I suggest, is an enhancement of guidance for a reader through the vast collection of evaluation steps done (see my first specific comment). This means a major revision (but likely not too high efforts) in terms of the paper presentation / structure, but only minor revisions in terms of its content.*

We would like to thank the Reviewer #3 for their positive feedback and specific comments which provided useful suggestions for improving our paper's readability. We coupled all revision comments in this sense, however trying to avoid making the paper too long.

*Specific comments*
*My main point is on the presentation / structure of the paper: The paper presents a rich set of analysis to assess the quality of the MONARCH reanalysis dataset. The authors have already made an effort to avoid overwhelming the reader by splitting of parts of the material into a supplement. However, even in the main paper as it is now a reader may easily loose the overview with so many different aspects. I therefore recommend for better guidance of the reader:*

*1) to provide an overview table in the beginning of section 2 or in section 2.6 which lists the different analysis made (e.g. satellite validation, reanalysis assessment of DOD, coarse DOD, annual, seasonal, …) versus the various reference datasets used; this should include the analysis shown in the supplement and at the same time identify the placing in the paper (main paper or supplement) for each analysis element*

We added an overview table in section 2.6, as the reviewer suggested. Thank you for the suggestion.

```
Revised text (lines: 367-370, Section 2.6):
```
The next two sections present the results of the MONARCH reanalysis assessment as well as the validation of the satellite datasets using ground-based measurements. An overview of the datasets evaluated and the datasets used as reference, of the spatial and temporal scales at which the evaluation was performed, and of the figures that depict the results is outlined in Table 1 to help the reader navigate between the following sections.

*2) to add sub or sub-sub headings (e.g. for the section on DOD and coarse DOD) in the analysis and discussion sections*

We did that. Thank you for the suggestion.

*3) to add short titles to the figures to help a quick reader*

We added titles to all the figures except for the MONARCH domain map (Fig. 1). Thank you for the suggestion.

*Further specific comments:*
*4) Some paragraphs do not seem to fit in the section they are presented in – they should be moved:*
*- lines 274 – 287: should better become an overview at the start of section 2 or in section 2.6 (and there the overview table which I suggested could be added, too)*
*- lines 347 – 357: should better be moved to the discussion section*

We changed the position of both paragraphs. The first was placed at the end of Section 2.5 to summarize the advantages that each observational dataset presented in the previous sections (2.2–2.5) brings to our study (lines: 314–327 of the revised manuscript).

The second paragraph was placed at the beginning of Section 4.2.1 (lines: 595–606 of the revised manuscript) to explain the possible reasons why the results of the reanalysis evaluation presented in Section 4.1.1 are not the same everywhere. On the contrary, in the regions where the differences between the observational datasets are mitigated and the evaluation scores are similar, the multi-sensor aggregation is favored, which is the subject of this section.

*5) In the abstract the tense is mixed up (mostly present, but at some places simple past is used, e.g. line 11) – this should be harmonized.*
*6) In the abstract (line 24/25) relative bias are provided as fractions of 1, which can easily be mixed up with absolute AOD values. I therefore recommend to use %-values.*

We fixed both of them. Thank you for the remarks.

*7) In the introduction impact of atmospheric dust is summarized; I miss a statement on fertilizing the Amazonas here.*

We added a statement as the reviewer suggested. Thank you for the suggestion.

`Revised text (lines: 43-44, Section 1):`
Once the dust is deposited, by wet or dry deposition, it impacts both aquatic and terrestrial ecological systems through their biogeochemistry, e.g., dust contains micronutrients that can act as a fertilizer increasing primary productivity in the Amazon rain forest (Okin et al., 2004; Jickells et al., 2005; Painter et al., 2007; Bristow et al., 2010; Lekunberri et al., 2010; Yu et al., 2015).

Bristow, C. S., Hudson-Edwards, K. A., and Chappell, A.: Fertilizing the Amazon and equatorial Atlantic with West African dust, Geophys. Res. Lett., 37, https://doi.org/10.1029/2010GL043486, 2010.

*8) In summarizing the MONARCH reanalysis in the introduction, data assimilation of specific satellite datasets is mentioned. Can you add here which datasets are used, so that a reader can understand immediately in how far these are different from the datasets which are used for the evaluation in this paper.*

Apart from the Introduction, in Section 2.1 of the submitted manuscript we provided further details about the dataset assimilated in the reanalysis; please note the lines 146–151 of the revised manuscript. Specifically, the assimilated data are coarse DOD retrievals obtained from MODIS Deep Blue (over-land) products, whereas the MIDAS dataset used for the evaluation was generated based on MODIS AOD data retrieved by merging products from different

algorithms, namely Dark Target (over-land and ocean) and Deep Blue (over-land). We revised the manuscript adding this information.

Revised text (lines: 170–171, Section 2.2):
MIDAS combines quality filtered AOD from MODIS Dark Target (over land and ocean) and Deep Blue (over land) products (NASA's Aqua satellite, Collection 6.1, Level 2; Sayer et al., 2014)  at swath level , along with DOD-to-AOD ratios provided by the Modern-Era Retrospective analysis for Research and Applications version 2 (MERRA-2) reanalysis …

Sayer, A. M., Munchak, L. A., Hsu, N. C., Levy, R. C., Bettenhausen, C., and Jeong, M.-J.: MODIS Collection 6 aerosol products: Com parison between Aqua's e-Deep Blue, Dark Target, and "merged" data sets, and usage recommendations, J. Geophys. Res.-Atmos., 119, 13,965–13,989, https://doi.org/10.1002/2014JD022453, 2014.

*9) Section 2.4: can you add a statement, why you chose this particular IASI dataset and what unique characteristics it has as compared to the IASI datasets provided by the operational Copernicus Climate Change Service / Climate Data Store (https://cds.climate.copernicus.eu/cdsapp#!/dataset/satellite-aerosol-properties?tab=overview).*

Unlike most IASI dust products (e.g., at Copernicus Climate Change Service), the AEROIASI dataset provides retrievals of both coarse DOD and dust extinction profiles that were used to assess the corresponding MONARCH reanalysis products. The present paper focuses on the evaluation of MONARCH DOD, and thus the comparison of the MONARCH reanalysis dust extinction profiles with AEROIASI is not included in this manuscript. Nevertheless, assessments of MONARCH coarse DOD with other IASI DOD retrievals (e.g., ULB) would also be useful and could be done in future work. The text was revised accordingly.

Revised text (lines: 234–235, Section 2.4):
Unlike most IASI dust products (e.g., Clarisse et al., 2019), the AEROIASI dataset provides both vertical and column-integrated dust extinction information. More specifically, AEROIASI products include twice-daily 3D distributions of dust extinction coefficient, although the present study only uses dust horizontal distributions derived in terms of DOD.

Clarisse, L., Clerbaux, C., Franco, B., Hadji-Lazaro, J., Whitburn, S., Kopp, A. K., Hurtmans, D., and Coheur, P.-F.: A decadal data set of global atmospheric dust retrieved from IASI satellite measurements, J. Geophys. Res-Atmos., 124, 1618–1647, https://doi.org/10.1029/2018JD029701, 2019.

*10) Figures 2 – 5: The top row images (2 per column) are too small to be able to see much in them (on paper) – please enlarge them both to the size of the other maps and place them one below the other on top of each column (this should still fit on one page)*

We modified the four figures as the reviewer suggested (see Figures 2–5 of the revised manuscript). Thank you for the suggestion.

*11) In many places you alternate "MONARCH" with "reanalysis" – for reader guidance I would find it easier to follow, if you keep "MONARCH" in all places*

We added the word "MONARCH" before the word "reanalysis" everywhere in the manuscript except where multiple repetitions of the expression "MONARCH reanalysis" in the same sentence/paragraph would make the text unreadable. MONARCH is the underlying model on which the reanalysis is based and to avoid misunderstandings it should not replace the word "reanalysis" in the text. However, for simplicity 's sake we use the expressions "MONARCH reanalysis DOD" and "MONARCH DOD" equivalently as we mention in the revised manuscript (lines: 166–167 of the revised manuscript, Section 2.1).

*12) Fig. 4 and 5: There is a strong land-sea contrast in the MB map for MIDAS, which I missed in the text – can you please add this*

We added a statement about this in the revised manuscript.

```
Revised text (lines: 489-490, Section 4.1.1):
```
Overall, the MONARCH reanalysis tends to underestimate DOD except in desert dust source regions where the reanalysis  and the observational datasets show some discrepancies. The comparison with MIDAS  reveals a strong MB discontinuity from land to sea and especially from the dust sources to the adjacent maritime regions (Fig. 4g). In particular, the MONARCH reanalysis shows overall overestimations in Northern Africa, …

```
Revised text (lines: 543-544, Section 4.1.2):
```
The MONARCH reanalysis overestimates the coarse DOD over all the dust sources when compared to MIDAS and AEROIASI (Fig. 5g-h) with values that exceed 0.1 over the Bodele Depression and its downwind areas as well as over the major dust sources of the western Sahara Desert. Again here, as in the case of DOD MB, a discontinuity in the MB for MIDAS is noted between land and oceans. The comparison against AERONET shows overall underestimations (MB < 0) with maxima (MB < -0.1) at stations situated downwind of the Bodélé Depression towards the Gulf of Guinea, in Cape Verde and close to the Registan Desert (Fig. 5i).

*Technical corrections*
*13) Line 52: delete "the" before "airborne aerosols"*
*14) Line 76: better replace "data availability" by "measurement possibility"*
*15) Line 79: better replace "e.g.," by "and" (as there are two separate aspects in this sentence)*

We corrected them all. Thank you for the remarks.

*16) Line 119/120: I do not understand the statement iii) – can you please extend or reword to explain what this means?*

We rephrased and extended this part in the revised manuscript.

```
Revised text (lines: 121-123, Section 2):
```
iii) datasets must be homogeneous – i.e., no changes in the algorithm's version or calibration of the instrument for the whole spatio-temporal domain – and harmonized – e.g., ground-based observations must be from international networks that implement a harmonized quality assurance and quality control procedure ;

*17) Line 131: better replace "-30° E" by "30° W"*
*18) Line 151: you could add "profile" before "variables"*

We revised them both. Thank you for the suggestions.

*19) Fig. 1: should the area "Northern Europe" not better be renamed to "Northern and Central Europe"?*

Based on the reviewer's suggestion we named the sub-region "North-Central Europe" and we replaced the old name in Fig. 1 and in the text. The acronym "NorEur" was kept the same.

*20) Line 175: can you add some quantitative information here (an average % value for example)*

We added it. Thank you for the suggestion.

Revised text (lines: 181–182, Section 2.2):
however, in terms of relative uncertainty the MIDAS DOD product is highly reliable over dust-rich regions (~ 33 % annual average in the regions with strongest DODs) and becomes more uncertain in areas where dust loading is infrequent.

*21) Lines 178 – 187: I find this discussion of technical matching in UTC confusing, as the difference in local time is smaller – maybe you should point this out in addition*

We simplified the text because the explanation was indeed somewhat confusing.

Revised text (lines: 188–190, Section 2.2):
Regarding the temporal collocation, thanks to the wide MODIS swath (~ 2330 km), MIDAS provides near-global DOD retrievals every 1 to 2 days; consequently, MONARCH 3-hourly time-steps had to be averaged around Aqua's overpass time. Aqua follows a sun-synchronous, near-polar orbit, crossing the Equator once during daytime at ~ 13:30 local time (LT) and. As MONARCH outputs are given in UTC, it was necessary to convert 13:30 LT to UTC units, which depends on longitude. The MONARCH spatial domain contains eight time-zones (15 degrees of longitude constitute one time-zone) from –2 hours to +5 hours, implying that Aqua/MODIS takes measurements over the MONARCH domain between 08:30 UTC (70° E) and 15:30 UTC (–30° E). Hhence, for a given longitude (assume 43° E) related to a certain time-zone (i.e., +3 hours), the MONARCH DOD was temporally averaged around that MODIS acquisition time (i.e., 10:30 UTC) using the two nearest MONARCH timeslots (i.e., 09:00 and 12:00 UTC).

Revised text (lines: 220–221, Section 2.3):
For the temporal collocation we followed a similar methodology as in the case of MIDAS: MISR on board of NASA's Terra satellite is crossing the equator on the descending node at about 10:30 LT. After having converted 10:30 LT to UTC time taking into account the related longitude, and the MONARCH DOD was temporally averaged around MISR overpass time using the two nearest MONARCH timeslots.

*22) Lines 244 and 260: the radius limits of 2 μm and 0.6 μm differ – can you please explain?*

In fact, there is a mistake in line 244, the value 2 μm refers to the diameter of the particles. The correct radius value is 1 μm and we have revised the text accordingly. Thank you for that!

   Regarding the radius limits, the AEROIASI coarse DOD dataset considers the contribution of all dust particles with radius larger than 0.6 μm (even if it is more sensitive to particles with r > 1 μm) because the a priori size distribution that was used as input to the AEROIASI algorithm is the average single-mode distribution for the coarse mode (r > 0.6 μm) of AERONET size distributions derived from Saharan stations.

Revised text (line: 249, Section 2.4):
Using thermal infrared measurements, AEROIASI retrievals are mostly sensitive to coarse aerosols (with a radius roughly greater than ~ 2 μm). In fact, the contribution of fine dust particles (with radii < ~ 1 μm) to total ADOD at 10 μm is expected to be less than ~ 10 % (Pierangelo et al., 2005); consequently, the AEROIASI product considered here is the coarse mode DOD at 10 μm.

*23) Line 282: nighttime measurements: are they used at all in this paper? Please state explicitly.*

Yes! IASI passes over the MONARCH domain twice a day, at around 09:30 and 21:30 local time, as it was mentioned in the submitted manuscript (line: 230 of the revised manuscript, Section 2.4). Consequently, the AEROIASI dataset used in this paper is the only one that contains data from nighttime measurements.

*24) Lines 284/285: "usually" and "in most cases" duplicate their meaning*

We fixed this by keeping "in most cases" (line: 325 of the revised manuscript, Section 2.5). Thank you for this remark!

*25) Lines 300/301: this means that these cases are not used at all, correct? Please state this explicitly*

Exactly! We revised the manuscript accordingly.

```
Revised text (lines: 291–292, Section 2.5):
```
Finally,  a mixed aerosol type is assumed when 0.75 ≤ AE ≤ 1.2 and since we cannot precisely estimate the contribution of dust to it, these cases are not used for the evaluation purposes of this study .

*26) Line 303: fine mode radius of 0.6 µm – you should mention / discuss the impact of the different definition elsewhere (0.5 µm; e.g., line 57)*

We changed the "i.e." to "e.g." in line 57 of the submitted manuscript (line: 58 of the revised manuscript, Section 1), because r > 0.5 µm is not a strict definition of coarse particle size, but an example of what that radius might be. In fact, only MIDAS uses 0.5 µm as a threshold radius whereas all the others coarse DOD datasets (MONARCH, AEROIASI and AERONET) use 0.6 µm. The potential impact that these definitions may have on the comparison between MONARCH and MIDAS coarse DOD, was mentioned in the submitted manuscript (lines: 710–712 of the revised manuscript, Section 4.2.2).

*27) Line 304: can you mention that wildfires may lead to significantly high AOD values (versus sea salt with only low values)?*

We mentioned it. Thank you for the suggestion.

```
Revised text (lines: 297–298, Section 2.5):
```
The coarse mode AOD is dominated by maritime/oceanic aerosols and desert dust, whereas other natural sources, such as wildfires, can also produce coarse-mode aerosols. Sea-salt is usually associated with low AOD (< 0.03) and mainly affects coastal stations, and therefore inland high coarse AOD values is assumed to be mineral dust, although significantly high AOD values could be associated with biomass burning particles because they are more absorbing than dust (Dubovik et al., 2002).

*28) Lines 317/318: I do not understand this sentence, can you please reword?*

We rephrased the sentence.

```
Revised text (lines: 309–310, Section 2.5):
```
The two datasets were spatially collocated by interpolating MONARCH over each AERONET station. Regarding the temporal collocation,  AERONET data are acquired at 15-minute intervals on average; therefore, all AERONET measurements within ±90 minutes of the MONARCH reanalysis  outputs have been averaged for the comparison on a 3-hourly basis.

*29) Line 320: can you please add the number of stations which meet this criterion?*

We added it. Thank you for the suggestion.

```
Revised text (line: 313, Section 2.5):
```
Figure 4u̶2̶ shows the location of the AERONET sites with at least 30 temporally collocated pairs available (224 in total).

*30) Lines 505 – 507: can you add a reference for this statement?*
We added them. Thank you for the suggestion.

```
Revised text (lines: 513-514, Section 4.1.1):
```
In MAM the meteorological conditions favor the transport of dust from the southern parts of the Sahara (e.g., Bodélé Depression) to the Sahel (DOD ~ 0.3 in SubSah; Kaly et al., 2015), and from the northern Sahara sources and the Syrian Desert (Fig. 1, "G") towards the Mediterranean (Solomos et al., 2018).

Kaly, F., Marticorena, B., Chatenet, B., Rajot, J., Janicot, S., Niang, A., Yahi, H., Thiria, S., Maman, A., Zakou, A., Coulibaly, B., Coulibaly, M., Koné, I., Traoré, S., Diallo, A., and Ndiaye, T.: Variability of mineral dust concentrations over West Africa monitored by the Sahelian Dust Transect, Atmos. Res., 164-165, 226–241, https://doi.org/10.1016/j.atmosres.2015.05.011, 2015.

Solomos, S., Kalivitis, N., Mihalopoulos, N., Amiridis, V., Kouvarakis, G., Gkikas, A., Binietoglou, I., Tsekeri, A., Kazadzis, S., Kottas, M., Pradhan, Y., Proestakis, E., Nastos, P. T., and Marenco, F.: From tropospheric folding to Khamsin and Foehn winds: How atmospheric dynamics advanced a record-breaking dust episode in Crete, Atmosphere, 9, https://doi.org/10.3390/atmos9070240, 2018.

*31) Line 709: please add "total column"*

We added it. Thank you for the suggestion.

```
Revised text (line: 748, Section 5):
```
Here in particular, we seek to assess the performance of the MONARCH reanalysis in reproducing the total column variables of total and coarse DOD using dust products derived from MODIS, MISR and IASI space-borne instruments along with ground-based remote-sensing measurements from AERONET.